# Social motility of biofilm-like microcolonies in a gliding bacterium

Chao Li [1,9], Amanda Hurley[2,3,9], Wei Hu[4], Jay W. Warrick[5], Gabriel L. Lozano[2,6], Jose M. Ayuso[1,5,7], Wenxiao Pan [4], Jo Handelsman[2,3] & David J. Beebe [1,5,8 ✉]

Bacterial biofilms are aggregates of surface-associated cells embedded in an extracellular polysaccharide (EPS) matrix, and are typically stationary. Studies of bacterial collective movement have largely focused on swarming motility mediated by flagella or pili, in the absence of a biofilm. Here, we describe a unique mode of collective movement by a self-propelled, surface-associated biofilm-like multicellular structure. *Flavobacterium johnsoniae* cells, which move by gliding motility, self-assemble into spherical microcolonies with EPS cores when observed by an under-oil open microfluidic system. Small microcolonies merge, creating larger ones. Microscopic analysis and computer simulation indicate that microcolonies move by cells at the base of the structure, attached to the surface by one pole of the cell. Biochemical and mutant analyses show that an active process drives microcolony self-assembly and motility, which depend on the bacterial gliding apparatus. We hypothesize that this mode of collective bacterial movement on solid surfaces may play potential roles in biofilm dynamics, bacterial cargo transport, or microbial adaptation. However, whether this collective motility occurs on plant roots or soil particles, the native environment for *F. johnsoniae*, is unknown.

[1] Carbone Cancer Center, University of Wisconsin-Madison, Madison, WI, USA. [2] Wisconsin Institute for Discovery, University of Wisconsin-Madison, Madison, WI, USA. [3] Department of Plant Pathology, University of Wisconsin-Madison, Madison, WI, USA. [4] Department of Mechanical Engineering, University of Wisconsin-Madison, Madison, WI, USA. [5] Department of Biomedical Engineering, University of Wisconsin-Madison, Madison, WI, USA. [6] Divisions of Infectious Diseases and Gastroenterology, Boston Children's Hospital and Harvard Medical School, Boston, MA, USA. [7] Morgridge Institute for Research, Madison, WI, USA. [8] Department of Pathology and Laboratory Medicine, University of Wisconsin-Madison, Madison, WI, USA. [9] These authors contributed equally: Chao Li, Amanda Hurley. ✉email: djbeebe@wisc.edu

Bacterial biofilms are consortia of surface-associated cells bound together with an EPS matrix[1–3]. Although individual bacteria can translocate across surfaces using flagella[4], type-IV pili[5,6], surfactants[7] or gliding motility apparati[8], biofilms are typically stationary. In some cases, however, biofilms have been reported to move. Small *Pseudomonas aeruginosa* microcolonies are dynamic, using type-IV pili to merge and split before settling down to form a sessile biofilm[9]. *Myxococcus xanthus* displays diverse multicellular and motile behaviors. Individual cells rely on gliding motility whereas social motility for bacterial predation and the formation of fruiting bodies primarily depends upon type-IV pili[10,11]. *Neisseria* species also use type-IV pili to form large microcolonies that move across a surface as discrete units and grow via merging[12,13]. In these examples, type-IV pili are important contributors to the movement of microcolonies. Type-IV pili function through the extension of pilin filaments from the cell body and attachment to the substrate[14]. The subsequent force of retraction and disassembly of the pilin subunit pulls the cell forward. The same principles apply to the motility of multicellular structures using type-IV pili[13].

Here, we describe a unique collective movement that does not involve pili or flagella, in the soil-dwelling bacterium *Flavobacterium johnsoniae*. Surface-associated, three-dimensional (3D) biofilm-like microcolonies form, move, merge and disperse within a 24-h period (Fig. 1). Individual *F. johnsoniae* cells move by gliding motility, which is critical during early aggregation of *F. johnsoniae* into microcolonies. Briefly, rod-shaped cells align parallel to the surface where a rotary gliding motor moves motility adhesins, such as SprB and RemA, around the cell on a helical track to generate cell movement[8]. By contrast, in the microcolony, cells contacting the surface at the base do not arrange horizontally but interact obliquely at a cell pole. The semi-vertical base cells suggest traditional gliding, in which cells are aligned in parallel with the surface to facilitate helical movement of adhesins, is not the primary translocation mechanism. However, perturbing gliding motor function either genetically or biochemically influences microcolony development, shape, and behavior. We conclude, therefore, that the rotary gliding motor plays a crucial role in microcolony motility.

## Results

### The structure of *F. johnsoniae* microcolonies.

The growth dynamics of *F. johnsoniae* and the translocation mechanism of the microcolonies were studied using a newly developed under-oil open microfluidic system (UOMS)[15,16] (Fig. 1). Bacterial cells were introduced into under-oil sessile microdrops on a glass substrate and monitored with bright-field time-lapse microscopy for 24 h (Methods). Microcolonies formed, migrated, merged, grew over time, and dispersed ~15 h after initiating culture growth (Supplementary Fig. 1, Supplementary Movie 1). To determine whether these structures could be biofilms, we stained them with fluorescently conjugated lectins to detect EPS. Only Concanavalin A (ConA), out of four lectins tested, specifically bound the *F. johnsoniae* microcolonies. As seen from the staining (Fig. 1d, e), ConA bound the core of a microcolony but not individual cells on the surface or planktonic cells, suggesting that EPS production is localized to the cells in microcolonies. Microcolonies bearing multiple, separate EPS cores were widely observed (Supplementary Movie 2), which is the result of multiple merging events. Indeed, ConA binding only demonstrates the presence of exposed polysaccharides, which could arise from capsular sugars and not biofilm matrix, but the absence of exterior staining of the microcolonies suggests microcolony-specific production and subsequent localization or perhaps a division of labor in which only interior, non-motile cells produce capsule.

### Microcolony motility and analysis of movement mechanism.

Time-lapse videos were converted to datasets using a custom particle tracking algorithm (Methods) to quantify the collective movement by distinguishing the features of microcolony behavior, including growing (or growing and moving), shrinking (or shrinking and moving), merging, and splitting (Fig. 2a, b, Supplementary Fig. 2). Specifically, the merging and splitting events were distinguished from the movement to remove the sudden increase of the shift of center-of-mass and thus a positive error in the particle tracking analysis. The resulting track displacement analysis (Fig. 2c) shows the scale of the microcolony movement. Compared to the active mechanism proposed where the single-cell movements in the microcolonies fuel microcolony movement, it's necessary to clarify whether the collective movement could be driven by Brownian motion, a common passive mechanism of particle diffusion. The mean squared displacement of particles in Brownian motion is inversely proportional to the particle size, but in the current system, the velocity correlates positively with microcolony size (Fig. 2f). In addition, the microcolony velocity decrease in the late stage (>15 h) with a diminishing microcolony size is due to the dispersal. Therefore, Brownian motion is not a dominant mechanism in the microcolony movement. To understand the mechanism of movement of microcolonies, we utilized mutant analysis, small molecule inhibition, and a computer simulation.

### Genetic analysis.

Since *F. johnsoniae* is a gliding bacterium, we first determined whether the gliding apparatus plays a role in the collective movement of microcolonies. Mutant analysis showed that both gliding and cell-cell interactions are required for normal microcolony development and motility. A mutant containing a transposon within *gldD*, which encodes a lipoprotein required for gliding and the type-IX protein secretion system (T9SS)[17,18], neither glides nor forms microcolonies (Fig. 2b–f, Supplementary Movie 3), revealing the dependence of microcolony formation on the gliding motility apparatus. To test this model, we analyzed a deletion mutant of the major motility adhesin, SprB, in which microcolony formation was completely abrogated. Similarly, a deletion mutant of the gene encoding the auxiliary motility adhesin, RemA, formed misshapen microcolonies defective in collective movement, suggesting gliding proteins are also required for mature microcolony movement (Supplementary Fig. 3). Gliding motility adhesins are not only important for surface attachment, but also for cell-cell interactions. Overexpression of RemA in *F. johnsoniae* results in strong autoaggregation in liquid culture through interaction with galactose-containing EPS on neighboring cells[19]. Whether or not those aggregates are motile is unknown but implicates the importance of gliding adhesins for cell-cell adherence within the microcolony and could explain the abnormal microcolony morphology in the *remA* mutant.

Since RemA has been suggested to mediate cell-cell interactions and *remA* mutants displayed defective microcolony motility, we explored the impact of other physical disruptions in *F. johnsoniae* collective motility. From a collection of *F. johnsoniae* transposon mutants, we selected *fjoh_0352*, which contains a mutation in a homolog of *wzz* and is predicted to construct the O-antigen of lipopolysaccharides (LPS). Mutations that affect O-antigen in *M. xanthus* reduce social motility[20] but, to our surprise, disruption of *fjoh_0352* had the opposite effect in *F. johnsoniae*. Mature *fjoh_0352* microcolonies exhibited a dramatic hyper-merging (Fig. 2b–d), hyper-motile (Fig. 2e, f) phenotype resulting in massive microcolony formation (Fig. 2d solid lines, Supplementary Movie 4). The increased motility of *fjoh_0352* can also be seen in the increased track displacement traveled by *fjoh_0352* compared to wild-type (WT) (Fig. 2c). We suggest that the absence of polar

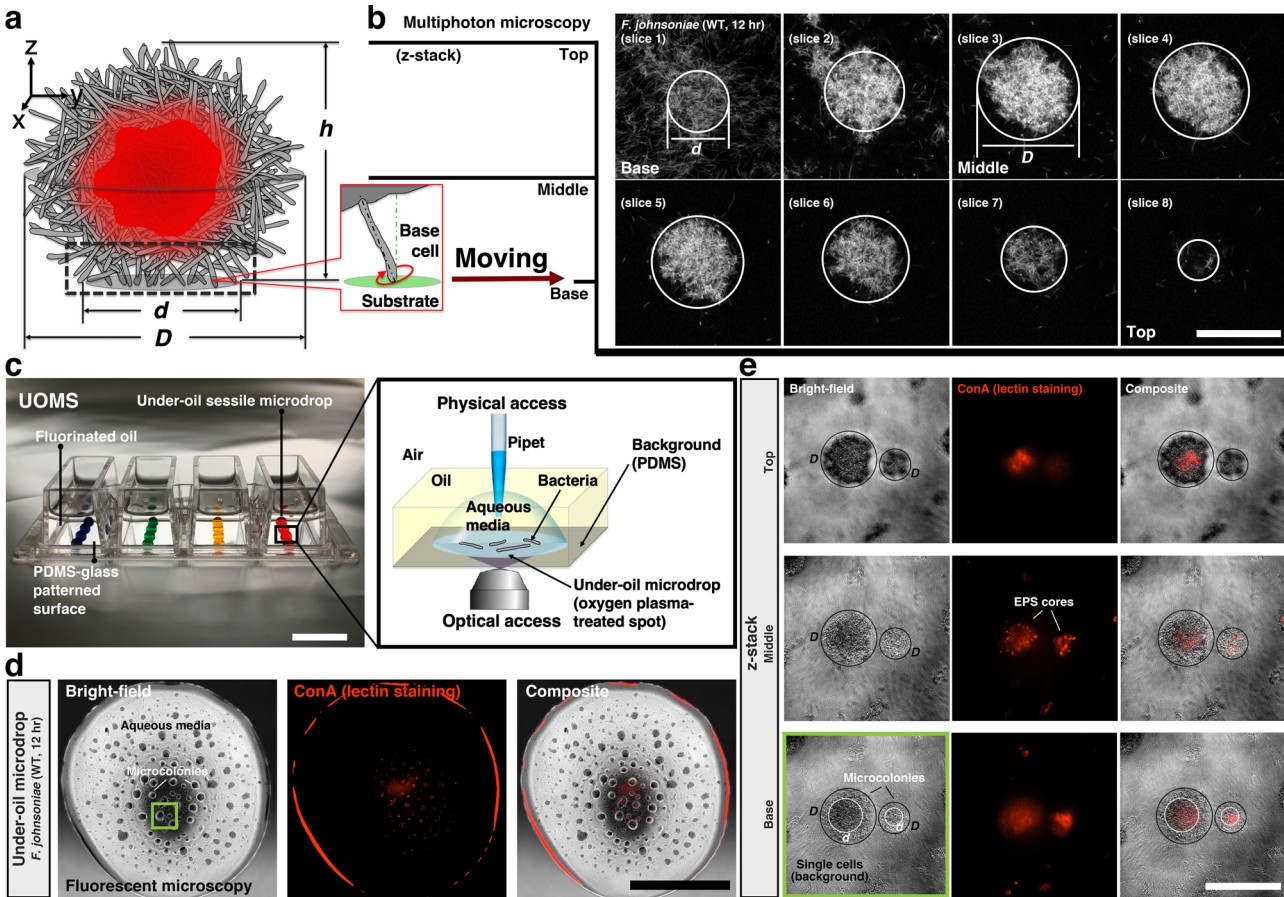

**Fig. 1 The microcolonies self-assembled from *F. johnsoniae* wild-type (WT). a** Schematic of a bacterial microcolony. The rod-like bacteria (gray) self-assemble into a microcolony (diameter, *D* and height, *h*), with extracellular polysaccharide (EPS) cores (red) and semi-vertical cells (or base cells, the dashed-line box) at the base (diameter, *d*). (Callout) The typical orientation of a single base cell relative to the substrate. **b** z-stack slices (8 total with 6 μm between slices) of a multiphoton microscopic image confirm the spherical, three-dimensional (3D) structure of a microcolony ($D \approx 60$ μm, $d \approx 36$ μm, $h \approx 45$ μm). The signal indicates reduced nicotinamide adenine dinucleotide (NADH) from cell metabolism. Scale bar, 50 μm. **c** The under-oil open microfluidic system (UOMS) with under-oil sessile microdrops for studying the growth dynamics of *F. johnsoniae*. Colored water (2 μl per spot, 2 mm in diameter) was used for visualization. PDMS represents polydimethylsiloxane. Scale bar, 1 cm. (Callout) Schematic of the under-oil environment around a sessile microdrop. **d** Microcolonies in a microdrop with fluorescent Concanavalin A (ConA) lectin staining (12 h after initiating culture growth). Scale bar, 1 mm. **e** z-stack high-magnification microscopic images of microcolonies in (**d** the green box). Distinct EPS cores (ConA+) enveloped by attached and actively moving cells can be detected within microcolonies that do not perfectly overlap with total cell mass. Scale bar, 100 μm. The experiments in **b**, **d**, and **e** were performed three times for the representative result. Source data are provided as a Source Data file.

oligosaccharides and exposure of lipid A of LPS increased hydrophobicity of cells in the *fjoh_3052* mutant, which in turn mediates stronger cell-cell interactions or binding within, and between, microcolonies. Furthermore, we found *fjoh_0352* exhibits greater autoaggregation compared to both WT and a *gldD* mutant, which could be the result of enhanced cell-cell interactions (Supplementary Fig. 4). Increased cell-cell interactions were similarly observed in *M. xanthus* mutants affected in the O-antigen[20]. Further genetic dissection and uncoupling of LPS, secreted polysaccharide pathways, and gliding will contribute to developing a model for cell-cell interactions within the motile microcolonies.

**Energy source of mature microcolony motility.** Genetic analysis revealed that gliding was necessary for the formation of microcolonies and contributed to their motility, suggesting the process is active and requires energy consumption. To test this possibility, we added the proton motive force (PMF) uncoupler carbonyl cyanide m-chlorophenylhydrazine (CCCP) to microdrops with pre-formed microcolonies (12-h pre-incubation). CCCP inhibits gliding motility of individual *F. johnsoniae* by interrupting the rotary movement of adhesins[8,21,22]. In the absence of CCCP,

mature microcolonies at 12 h and 15 h display increased size concurrent with a reduction of total number over time, demonstrating the ability of the microcolonies to migrate towards one another and merge to form larger, but fewer, microcolonies (Fig. 3). Both WT (Fig. 3b columns 1 and 2) and the mutant, *fjoh_0352* (Fig. 3b columns 5 and 6), exhibited this behavior, albeit *fjoh_0352* microcolonies were nearly twice the size as WT at 15 h, further supporting a hyper-merging phenotype (Fig. 3b median diameter 62 μm vs 114 μm, respectively). In the presence of CCCP, the WT microcolony growth was abrogated (Fig. 3b columns 3 and 4, Supplementary Movie 5, Supplementary Movie 6), whereas the overall abundance of microcolonies was increased (Fig. 3b columns 2 and 4), suggesting normal merging and motility requires the PMF and, thus, gliding. While not completely abolished, the *fjoh_0352* microcolony growth from 12 to 15 h was reduced in the presence of CCCP (Fig. 3b columns 7 and 8, Supplementary Movie 7, Supplementary Movie 8) compared to growth in its absence (Fig. 3b columns 5 and 6) and the overall abundance of microcolonies was increased (Fig. 3b columns 6 and 8), similar to WT. Surprisingly, CCCP did not fully inhibit the ability of *fjoh_0352* to merge. Incomplete inhibition

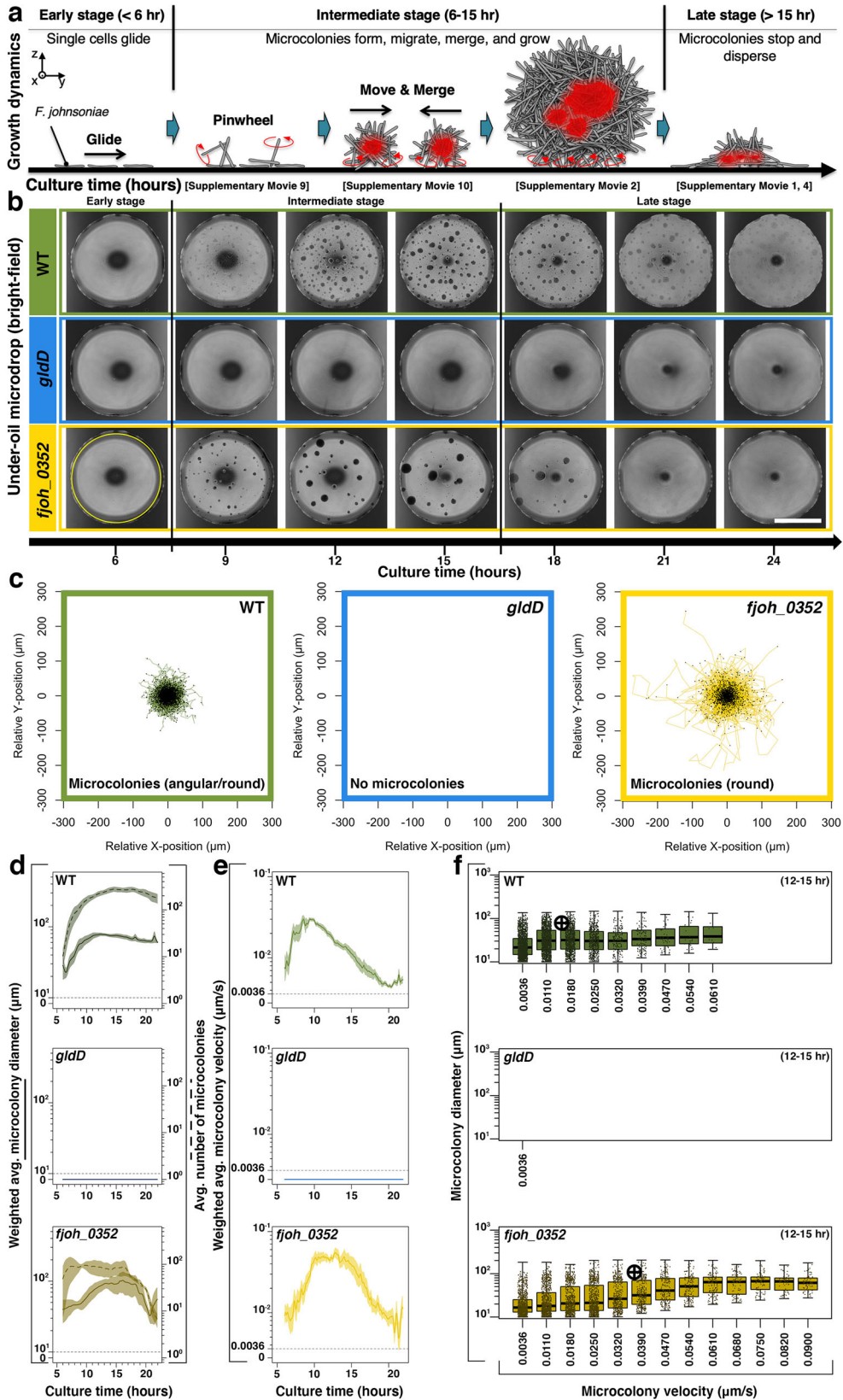

could be caused by lack of CCCP penetration into large microcolonies or into the mutant cells themselves, or perhaps a secondary mechanism contributes to the motility of microcolonies. In any case, chains of microcolonies were found in the presence of CCCP for both WT and *fjoh_0352* microdrops (Fig. 3a small red boxes), hinting that the PMF-driven gliding apparatus may be

required for movement of two microcolonies towards each other as well as the actual merging process itself.

**Surface orientation of base cells and behavior in the microcolonies.** We next investigated how individual cell motility fuels

**Fig. 2 Growth dynamics and microcolony motility. a** Schematic shows typical stages of the growth dynamics. In the early stage (< 6 h after initiating culture growth), the bacteria remain mostly as single cells and glide with minimal aggregation. In the intermediate stage (6–15 h), microcolonies start to form, migrate, merge, and grow. In the late stage (>15 h), microcolonies stop moving and disperse into single cells. **b** Comparison matrix with bright-field microscopic images between wild-type (WT) (green), *gldD* (blue), and *fjoh_0352* (yellow) over time. Scale bar, 1 mm. The experiments were performed three times for the representative result. **c** Spider plots of the track displacement from 6 to 21 h. A region of interest (ROI) (1.84 mm in diameter, the yellow solid line circle in **b** was set in particle tracking to avoid artifacts from the edge of the microdrop. All of the starting points of the movement of microcolonies converged to the origin (i.e., $x = y = 0\ \mu m$). **d** Weighted average microcolony diameter (solid line) and average number of microcolonies (dashed line) versus culture time (step, 5 min). **e** Weighted average microcolony velocity versus culture time (step, 5 min). Estimated microcolony volumes were used as weights for all weighted average calculations. The standard error is represented by the envelope on the plots in **d** and **e**. **f** Strip plots show the distribution of microcolony diameter versus microcolony velocity at the growth window (12–15 h). Data points were pooled from three biological replicates. Crosses indicate medians. Box edges and center represent the 25th, 75th, and 50th percentile while whiskers represent 1.5 × IQR (interquartile range) or the minima and maxima (whichever is less). Source data are provided as a Source Data file.

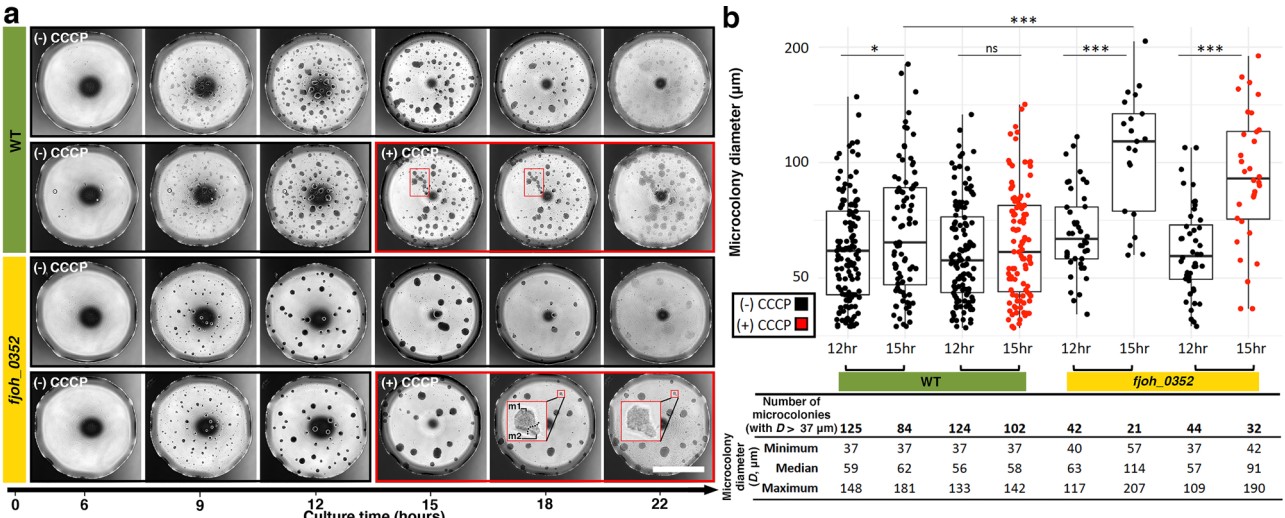

**Fig. 3 Proton motive force (PMF) uncoupler shows microcolony motility and merging is an active process. a** Comparison matrix with bright-field microscopic images between wild-type (WT) and *fjoh_0352* over time. A single biological replicate is shown in the matrix, representative of three biological replicates. Immediately after the 12 h time point, 10 μM of carbonyl cyanide m-chlorophenylhydrazine (CCCP) was added to the microdrops, indicated by the large red box, with microdrops in the absence of CCCP as a control. CCCP inhibits PMF and causes microcolonies to freeze before merging, highlighted by the small red boxes. Scale bar, 1 mm. **b** Microcolony size (diameter, *D*) and abundance as a function of time, *F. johnsoniae* strain, and presence (+)/absence (−) of CCCP of the biological replicate shown in **a**. Only mature microcolonies (*D* > 37 μm) were quantified to measure merging and motility rather than formation of new microcolonies. Microdrops with (+) CCCP are displayed in red. Box edges and center represent the 25th, 75th, and 50th percentile while whiskers represent the minima and maxima. Significance was determined by a non-parametric, two-sided *t*-test: *$P = 0.0469$, ***$P < 1.8e−5$, and ns non-significant. Source data are provided as a Source Data file.

microcolony movement. An intriguing possibility is that microcolony translocation occurs via cargo transport of EPS-bound cells in a manner similar to that displayed by *Capnocytophaga gingivalis*, a gliding relative of *F. johnsoniae* found in the human oral microbiome[23]. To explore this possibility, we examined cell morphology and movement on a surface in different growth stages. If cargo transport based on gliding motility were the mechanism, we would expect to see cells arranged parallel with the surface at the base of the microcolony[8]. This orientation was indeed observed among individual *F. johnsoniae* cells, but in the early-stage microcolony formation, the cells became oblique to the surface upon contact with each other, showing a pinwheel movement that was also previously reported in *F. johnsoniae* (i.e., spinning along an axis with a tethered pole)[21,24] (Fig. 4a, Supplementary Movie 9). The pinwheeling cells appear as rods with a changing length when projecting into the focal plane or round when perpendicular to it[25]. Analysis of cell shape at the base of mature microcolonies revealed round objects (Fig. 4b, c, Supplementary Fig. 5a), rather than cylindrical, indicating the cells were semi-vertical to the surface. We refer to these cells as base cells. It is significant that the base cells come in and out of the focal plane under microcolonies at about 2 Hz (Supplementary

Movie 10), suggesting base cells also exhibit pinwheel movement. Previous studies have described the pinwheel movement of *Flavobacterium* cells, involving the tethering of the rotary gliding apparatus, to occur both on a surface and on adjacent cells[21]. Whether or not the observed pinwheel movement is involved with force generation remains to be determined.

**Computer simulation of the drag forces on base cell and microcolony.** We next sought to determine how the base cells contribute microcolony motility. To generate microcolony motility, the net effect of the movement from individual base cells must be a non-zero value. Otherwise, the composition of the forces from the base cells would be canceled out, leaving the surface-associated microcolony non-motile. Regardless of the specific translocation mechanism of individual base cells, we aimed to quantify the net force percentage (NFP) of base cells to propel microcolony movement. We define the quantitative variable "NFP" as the equivalent percentage of drag force contributed by the base cells (i.e., neglecting the heterogeneity of force outputs among individual base cells) to match the drag force required to propel the microcolony at a given velocity.

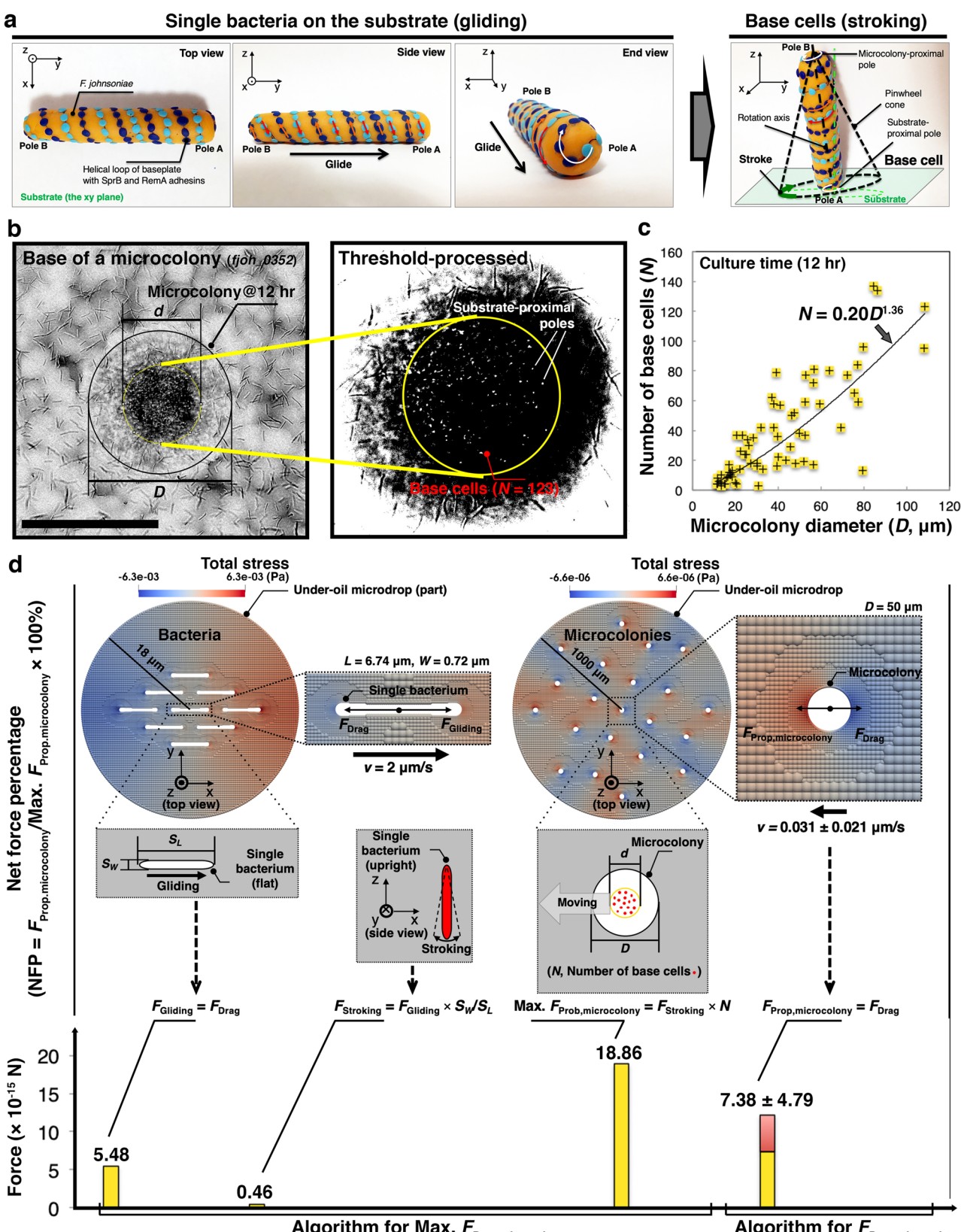

Typically to quantify the drag force of a motile particle (e.g., a cell or a bacterium), one would use the Stokes law whose prediction describes the relationship between the velocity of a particle in a fluid and the total stress exerted on the particle by the fluid. However, the Stokes law is valid only if we consider an isolated object moving in an unbounded (infinitely large) fluid environment; otherwise, the boundary effect and the hydrodynamic interaction between the object and other surrounding objects must be considered. In the case of *F. johnsoniae*, the movement of single cells and microcolonies involved multiple moving objects and thus would not be accurately captured by the Stokes law alone. Additionally, the Stokes law assumes an infinite,

**Fig. 4 Base cell contribution to microcolony motility. a** A clay model elaborates the potential base cell mechanism of movement based on a proposed transition in orientation and mode of movement from a single bacterium gliding on substrate to a base cell engaging in pinwheel movement at the base of a microcolony. The angle between the rotation axis of the base cell (the black dash-dot line) and the normal of the substrate (the green dash-dot line) could convert the pinwheel movement to a repetitive tapping contact movement, defined as stroking. It's worth noting the simulation is not limited to this specific mechanism. **b** A microscopic image (z direction) shows a typical microcolony from the hyper-motile mutant *fjoh_0352* (the black circle). The focal plane was set on cells at the base of the microcolony (the yellow circle). (Callout) Threshold-processed, amplified image of substrate-proximal poles (i.e., the bright dots) of the base cells. Scale bar, 100 μm. The experiments were performed three times for the representative result. **c** Number of base cells (*N*, the bright dots in the callout of **b** and later shown as red dots in **d**) selected by ImageJ (Supplementary Fig. 5a) as a function of microcolony diameter (*D*) (12 h after initiating culture growth). Microcolonies with *D* varying from ~10 to ~100 μm were counted. The solid line shows the power fitting, revealing a nonlinear relationship. **d** Net force percentage (NFP) of base cells obtained by computational simulations. Two algorithms were established to evaluate and compare the net propulsion force on a microcolony ($F_{Prop,microcolony}$) and its theoretical maximum (Max. $F_{Prop,microcolony}$) with NFP defined as $F_{Prop,microcolony}$/Max. $F_{Prop,microcolony}$ × 100%. First, the stroking force of a base cell ($F_{Stroking}$) was determined from the gliding force ($F_{Gliding}$) of *F. johnsoniae* based on the area in contact with the surface, i.e., $S_L = L \times W$ for a cell lying flat shown in white, $S_W = \pi (W/2)^2$ for a cell perpendicular to the surface shown in red, where *L* and *W* are the cell length and width, respectively. Gliding bacteria were arranged parallel to each other with an experimental density in a cropped microdrop (*r* = 18 μm in radius), moving in the same direction (from left to right) at a velocity of *v* = 2 μm/s. $F_{Stroking}$ was evaluated for the bacterium at the center of a microdrop. Max. $F_{Prop,microcolony}$ was then determined as $F_{Stroking} \times N$, corresponding to 100% NFP, i.e., all base cells generate force in the same direction. Second, $F_{Prop,microcolony}$ was estimated from the measured microcolony velocity. microcolonies (50 μm in diameter, *N* = 41 obtained in **c**) were arranged randomly with an experimental density across a microdrop, moving in random directions at an average velocity of *v* = 0.031 μm/s. $F_{Prop,microcolony}$ was estimated for the microcolony at the center of the microdrop. Here, the average velocity was used to reflect the average level of base cell participation in force generation. Given the low Reynolds number (Re ≈ 1e−6 in this study), $F_{Gliding}$ or $F_{Prop,microcolony}$ is balanced with the drag force ($F_{Drag}$) exerted by the fluidic environment[52], and hence can be evaluated from $F_{Drag}$. The color contour (unit: Pa) denotes the distribution of total stress in the fluid around the objects solved numerically by generalized moving least squares (GMLS) with an adaptive resolution in the discretization (Methods). Error bars, mean (the yellow bar) ± s.d. (the red bar). The simulated results were obtained based on the averaged input data from three biological replicates. Source data are provided as a Source Data file.

boundaryless environment, whereas the water:oil:surface interface of the microdrop creates a pressure boundary, which contradicts this assumption. To overcome these barriers of the Stokes law and increase precision, we conducted a computational simulation for a microcolony of a given size and velocity. We employed a numerical method based on the generalized moving least squares (GMLS) discretization[26–28] to solve the full Stokes equations (see SI). The GMLS discretization is built upon a rigorous approximation theory[27] and can be considered as a generalization of the finite difference method to unstructured grids while ensuring arbitrarily high order accurate and stable solutions of the Stokes equations[26,28]. Augmented with the spatially adaptive refinement technique, it has been shown to achieve optimal numerical convergence and computational efficiency[26] (Supplementary Fig. 6). Thus the GMLS method provides a high-fidelity tool for simulating the movement of particles in a low Reynolds number fluidic environment. The Stokes equations and the computational model in this study were established based on the experimental setup and the premises as follows: (i) The dynamics of the system lies in the low Reynolds number regime (Re ≈ 1e−6) considering the small inertia effect produced by the small length scale and velocity of the bacteria and microcolonies. Therefore, the propulsion force is always balanced with the drag force ($F_{Drag}$) exerted by the fluid. (ii) The simulation model was reduced to two-dimensional (2D) since the movements of bacteria and microcolonies are all surface-associated (iii).

The under-oil microdrop is treated as an incompressible fluid with a finite boundary. (iv) The possible rotation of the bacteria or microcolonies is negligible and hence not considered in the simulation. (v) All the input data used in the simulation was from the experimental results of the hyper-motile mutant *fjoh_0352*.

To probe NFP of the base cells, we performed two sets of numerical calculations. In the first set (Fig. 4d the left panel), we determined the propulsion force from the gliding of a single bacterium ($F_{Gliding}$). The target bacterium was immersed in a crowd of bacteria (×8, set to move in parallel in one direction based on experimental observation) with a density set to recapitulate the experimental condition. The maximum velocity of *F. johnsoniae* by gliding is about 2 μm/s, as reported in literature[8]. With this velocity,

we computed the corresponding $F_{Drag}$ on the target bacterium at the center of the microdrop. By force balance, we could determine $F_{Gliding}$; i.e., $F_{Gliding} = F_{Drag} = 5.48 \times 10^{-15}$ N. Considering the gliding mechanism of *F. johnsoniae* with the motility adhesins, we assume $F_{Gliding}$ is proportional to the contact area between the bacterium and the substrate surface. Thus, by comparing the contact area of a bacterium when gliding in parallel to the surface or stroking as a base cell interacting at its pole, we estimated the propulsion force generated by a base cell in stroking as: $F_{Stroking} = F_{Gliding} \times S_W / S_L = 0.46 \times 10^{-15}$ N, where $S_L = L \times W$ for a cell gliding in parallel on the surface, $S_W = \pi (W/2)^2$ for a base cell poised obliquely to the surface, *L* and *W* are the average cell length and width of the bacteria obtained from the ctFIRE analysis (Supplementary Fig. 1). In the second set (Fig. 4d the right panel), we started from computing $F_{Drag}$ on a microcolony. Within a capacity of calculation, we deployed multiple microcolonies (×21, set to move with the same velocity but random direction of the initial movement based on experimental observation) in an under-oil microdrop. The diameters of the microcolonies and the microdrop are set as 50 μm and 2 mm, respectively. The experimentally measured average velocity of the (*fjoh_0352*) microcolonies with the average diameter of 50 μm is 0.031 ± 0.021 μm/s. We computed $F_{Drag}$ exerted on the target microcolony at the center of the microdrop. Based on the force balance, the net propulsion force on a microcolony can be obtained as: $F_{Prop,microcolony} = F_{Drag} = 7.38 \pm 4.79 \times 10^{-15}$ N. Based on the observation that there are an average of 41 base cells in a microcolony with the diameter of 50 μm (Fig. 4c), we estimated the theoretical maximum propulsion force in a microcolony as: Max. $F_{Prop,microcolony} = F_{Stroking} \times N = 18.86 \times 10^{-15}$ N, assuming that all base cells contribute equally and generate force in the same direction of the microcolony movement, i.e., 100% NFP. Then, NFP can be obtained as $F_{Prop,microcolony}$/Max. $F_{Prop,microcolony}$ × 100% (e.g., 39.1% ±25.4% for the 50 μm microcolonies). The simulation results predicted an average NFP of 34% in microcolonies up to 100 μm in diameter, which suggests at least an equivalence of one third of the base cells acting together in force generation to propel microcolony motility at the measured velocity (Supplementary Fig. 7). For a small group of microcolonies that show a velocity as high as 0.075 μm/s, the NFP can spike to about 91%. While the NFP may be the result of

random base cell movement, current data do not rule out higher-order coordination or regulation among the base cells.

## Discussion

In this study, we report the discovery of a mode of collective movement of bacteria. *F. johnsoniae*, a non-flagellated, non-piliated bacterium, self-assembles into 3D structures with EPS cores, exhibiting translational movement across surfaces on semi-vertical, not horizontal, base cells. The closest structural relative of the *F. johnsoniae* collective motility appears in *Neisseria* species and are called microcolonies. However, due to the basis of movement and orientation of base cells, we designate this subset of microcolonies as "zorbs" to capture their dimensionality and the appearance of being moved by cells that reflect the legs on a zorb. We propose that the zorbs are biofilms, based on the cohesive multicellular structure and localization of EPS as detected by ConA binding (Fig. 1e, Supplementary Movie 2). In fact, localized ConA staining at the core of a biofilm has been seen previously in *Pseudomonas syringae*, another soil bacterium[29], in which ConA binds levan, a branched, fructan polysaccharide. Levan is an intriguing candidate for the matrix of a motile biofilm, as it is water soluble and remains fluid in solution, perhaps contributing flexibility as well as structural integrity[30,31]. Comparison of annotated levansucrase genes (GO:0050053) from other species of *Flavobacterium* to the *F. johnsoniae* UW101 genome identified *fjoh_2883* (65-69% identity) as a possible levansucrase-encoding homolog. A combination of polysaccharide chemical analysis, mutant analysis of *fjoh_2883* and biochemical intervention in WT zorbs with polymer hydrolyzing enzymes will determine matrix components of the motile *F. johnsoniae* zorbs. Such studies could illuminate tradeoffs between structural stability, flexibility, merging, and dispersal of *F. johnsoniae* motile zorbs.

Collective motility of gliding bacteria has been studied most thoroughly in *Myxococcus*, which coordinates rotary gliding motors and type-IV pili to form predatory swarms and fruiting bodies[6,32,33]. In addition, a relative of *F. johnsoniae*, *Capnocytophaga gingivalis*, exhibits "cargo transport," a behavior in which *C. gingaivalis* transports aggregates of non-motile species within the oral microbiome across a surface via gliding[23]. Although cargo transport might explain motility of zorbs, it is not supported by the data showing that *F. johnsoniae* cells at the base of zorbs do not align horizontally with the surface as do *C. gingivalis* cells during cargo transport. By contrast, the cells within motile zorbs are dynamic and the majority seem to interact with the surface by their poles (Fig. 4b, Supplementary Movie 10). This collective motility is not due to Brownian motion but rather is an active process as the motility increases with size of the zorb (Fig. 2f) and is dependent on PMF (Fig. 3). The direction of the motility appeared random, similar to locomotion seen in immune cells[34–36], although further work will determine whether directionality can be induced through chemical signals, which could provide clues to the function of zorb movement.

Collective movement and merging of microcolonies on this scale, to our knowledge, has only been reported previously in *Neisseria* species[37,38]. Although the mechanism of *Neisseria* colony translocation differs from *F. johnsoniae* as *Neisseria* movement depends on pili[12], the phenotypic parallel suggests evolutionary convergence and biological relevance of social motility. While zorb development could be caused by stresses generated by the under-oil microenvironment, and thus could be an environment-dependent adaptation, the discovery of motile microcolonies generated by a gliding bacterium expands our understanding of prokaryotic collective behavior.

How the activity of gliding apparatus from individual *F. johnsoniae* cells generates the movement of an entire zorb remains unknown. The movement of the zorbs via base cells

briefly gliding when attached to the surface has not been conclusively excluded. Additionally, the many base cells may act as focal adhesions successively binding to the substrate, not unlike the pili used by *Neisseria*. However, the observations of pinwheeling and the dynamic nature of the cells within the zorb could signify a "power stroke" force originating from base cells tethered by their rotary gliding motors that propel the zorb forward upon contact with the surface. This power stroke could originate from cells tethered either to the surface or the zorb itself. It is our assumption that individual cell forces are equal to the drag force, which requires further clarification in the future. Agnostic to mechanism, the computational simulations overcome limitations and assumptions of the Stokes law to determine more precisely that at least the equivalent of one third of the base cells act simultaneously to propel the collective group. Future experiments with fluorescent WT cells to track behavior of single cells within the zorb, especially at the base, will begin to differentiate among the possible mechanisms.

In other systems, microcolony translocation is important for virulence, specifically in *N. meningiditis* and *N. gonorrheae*[37,38], providing tantalizing evidence for the connection of social motility and host health. *F. johnsoniae* is a common rhizosphere resident, and whether this collective motility occurs on plant roots or soil particles, the native environment for *F. johnsoniae*, remains to be observed. Future work to determine the chemical and physical signals regulating zorb formation and motility could reveal environmental conditions favorable for this type of social motility and possible relevance for host-microbe interactions. Understanding the mechanisms of regulation could enable manipulation of the structure, and thus behavior, of *F. johnsoniae* and other gliding bacteria. Manipulating bacterial collective movement may contribute to strategies that improve predictable manipulation of microbial communities. The collective forces that translocate *F. johnsoniae* zorbs may contribute to areas beyond microbiology including designing functional soft materials[39], cargo transport strategies[23], and soft robotic systems[40] with dynamic self-assembly[41].

## Methods

**Preparation of the under-oil open microfluidic system (UOMS) device.** Chambered coverglass (4-well, #1.0 Borosilicate Glass, 0.13–0.17 mm thick, Fisher Scientific, 155383) was treated first with oxygen plasma (Diener Electronic Femto, Plasma Surface Technology) at 60 W for 3 min then moved to a vacuum desiccator (Bel-Art F420220000, Fisher Scientific, 08-594-16B) for surface modification via vapor phase deposition. Polydimethylsiloxane (PDMS)-silane (1,3-dichlorotetramethylsiloxane, Gelest, SID3372.0) (about 10 µl per device) was vaporized under reduced pressure for 3 min and then condensed onto glass substrate under vacuum at room temperature for 1 h. The PDMS-grafted surface was thoroughly rinsed with ethanol (anhydrous, 99.5%, Fisher Scientific) and deionized water and then dried with nitrogen. A custom silicone rubber mask (press-to-seal, 0.5 mm thick, with holes 2 mm in diameter, Grace Bio-Labs) was applied to the PDMS-grafted surface. The masked surface was then treated again with oxygen plasma at 60 W for 1 min. The PDMS moieties from the areas (or the spots) exposed to oxygen plasma were removed or oxidized, thus introducing surface chemistry contrast (i.e., oxygen plasma-treated spots versus PDMS on the background) defined by the pattern transferred from the mask to the surface. After surface patterning, the mask was removed by tweezers. Fluorinated oil (Fluorinert FC-40, Sigma Aldrich) was added to the wells by pipette (300 µl per well) for use in bacterial culture (Fig. 1c). Detailed discussions of the physics of surface patterning can be found in previous publications[15,16].

**Bacterial strains and growth conditions.** *F. johnsoniae* UW101 was the wild-type (WT) strain used in this study (Supplementary Table 1). Mutations in *gldD* (*fjoh_1540*) and *fjoh_0352* were generated using the pSAM-*Fjoh2* transposon (see Methods below) and maintained with 100 µg/ml erythromycin. Both *gldD* and *fjoh_0352* are located at the end of short operons (3 and 5 genes, respectively), so polar effects of the transposon on neighboring genes are unlikely. The transposon in *gldD* inserted at base pair (bp) 420 out of 561 total bp while the transposon in *fjoh_0352* inserted at bp 661 out of 1074 total bp, introducing a frameshift and stop codon after 226 amino acids. The *sprB* and *remA* mutants were provided by the McBride Lab at the University of Wisconsin-Milwaukee, which were recovered on

Lysogeny Broth (LB) with 100 µg/ml streptomycin and 20 µg/ml tetracycline to maintain pRR39-RemA, if necessary. Single colonies on LB agar plates with appropriate antibiotics were picked for overnight growth in $0.5 \times$ TBS at 28 °C without antibiotics, except for CJ2005, which was grown in 20 µg/ml tetracycline. Cultures were then washed twice with 10 mM sodium chloride and diluted in $0.1 \times$ TBS to optical density at the wavelength of 600 nm ($OD_{600}$) = 0.008 or ~$10^7$ colony-forming units (CFU)/ml. Microdrops of 2 µl for each strain were inoculated under oil by pipette and then cultured and monitored on microscope (Nikon Eclipse Ti) in time lapse at room temperature. For experiments containing carbonyl cyanide m-chlorophenylhydrazine (CCCP), 0.2 µl of a 110 µM CCCP stock solution in $0.1 \times$ TBS was manually added to microdrops after 12 h of incubation for a final concentration of 10 µM. The same volume of $0.1 \times$ TBS was added to control microdrops.

**Construction of pSAM_Fjoh2 vector.** The transposon delivery vector, pSAM_F-joh2 was derived from pSAM-BT[42]. The erythromycin resistance gene was removed with XhoI and XbaI (New England Biolabs, Ipswich, MA) and replaced with a different erythromycin resistance gene that was amplified by polymerase chain reaction (PCR) with primers ermF5_XhoI and ermF3_XbaI from pCP29[43] (Supplementary Table 2). The transposase gene was removed with BamHI and NotI (New England Biolabs) and replaced with the same transposon fused with the promoter of *F. johnsoniae rpoD* (fjoh_1433). Specifically, three hundred bp upstream of the start codon of fjoh_1433 were amplified using fjoh_1433_BamHI and fjoh_1433_TransRev, and the transposase was amplified using fjoh_1433_-TransFor and Transpo_Rev from pSAM-BT20. These PCR products were combined by a modified version of overlap extension (OE) PCR, cloned into pENTR/D-TOPO, and recovered using BamHI and NotI.

**Lectin staining of extracellular polysaccharide (EPS) matrix.** Lectin dyes were used to detect the EPS matrix in the *F. johnsoniae* microcolonies. Concanavalin A [ConA, Excitation/Emission (Ex/Em) at 555/580 nm, Fisher Scientific, C860] selectively binds to α-mannopyranosyl and α-glucopyranosyl residues. Lectin phytohemagglutinin-L (PHA-L, Ex/Em at 590/617 nm, Fisher Scientific, L32456) binds to N-acetylglucosamine β(1-2) mannopyranosyl residues. Lectin soybean agglutinin (SBA, Ex/Em at 650/668 nm, Fisher Scientific, L32463) selectively binds terminal α- and β-N-acetylgalactosamine and galactopyranosyl residues. Wheat germ agglutinin (WGA, Ex/Em at 496/524 nm, Fisher Scientific, W6748) binds to sialic acid and N-acetylglucosaminyl residues. After 12 h of growth, 0.2 µl of ConA, PHA-L, SBA, or WBA was added to the microdrop for a final concentration of 0.005, 0.05, 0.05, or 0.1 mg/ml, respectively. Samples were stained for 30 min and then the images were recorded (Nikon Eclipse Ti) without washing. Only ConA successfully stained the microcolonies (Fig. 1d, e and Supplementary Movie 2).

**Imaging and time lapse.** Nikon Eclipse Ti (4× and 40× objectives with 1.5× tube lens) was used to acquire the bright-field and fluorescent images with normalized lookup tables (LUTs), and run the time lapse (5 min interval). A multiphoton microscope (Bruker Fluorescence Microscopy, Middleton, WI) was used to acquire the reduced nicotinamide adenine dinucleotide (NADH) autofluorescence intensity images (Fig. 1b)[44,45]. The multiphoton microscope comprises a titanium:sapphire laser (Spectra Physics, Insight DS-Dual) and a 40× water immersion (1.15NA, Nikon) objective and temperature/carbon dioxide ($CO_2$) control system to keep the conditions at 28 °C and 5% respectively. NADH images were collected using an excitation wavelength of 750 nm and an emission bandpass filter of 440/80 nm. Pixel dwell time was set to 4.8 µs to acquire 1024 × 1024 pixel images, and the average of four images was calculated for each field of view. Subsequently, fluorescence lifetime images were collected using time-correlated single-photon counting electronics (SPC-150, Becker and Hickl) and a GaAsP photomultiplier tube (H7422P-40, Hamamatsu). 512-pixel images were obtained using a pixel dwell time of 4.8 µs over 60 s total integration time. The instrument response function was calculated from the second harmonic generation of urea crystals excited at 900 nm, and the full width at half maximum (FWHM) was measured to be 244 ps. A Fluoresbrite YG microsphere (Polysciences Inc.) was imaged as a daily standard for fluorescence lifetime. To avoid photobleaching and phototoxicity, the photon count rates were maintained at $1–2 \times 10^5$ photons/sec.

**Single-cell organization analysis.** The evolution of single-cell organization on the surface and in microcolonies was analyzed using the ctFIRE program [CT-FIRE for Individual Fiber Extraction (Current Version: V2.0 Beta), https://loci.wisc.edu/software/ctfire] (Supplementary Fig. 1). Cell size (i.e., length and width), straightness, and location of the rod-like *F. johnsoniae* in culture at different time points can be obtained directly from the output of ctFIRE. We used the output to further calculate the orientation of each bacterium in R/RStudio. For each bacterium, we identified all neighboring bacteria within a search radius of 9.8 µm. We then calculated the relative angle between the bacterium of interest and those neighbors and then used those to calculate a weighted mean. The weighting was based on distance to the neighbor (weight = 1 − radius/search radius). We then plotted the histogram of these average relative angles. Note that ctFIRE is a powerful image analysis tool for identifying and quantifying objects in a fiber-shaped morphology (i.e., large aspect ratio). In comparison, ctFIRE doesn't

efficiently capture circular particles or dots, e.g., the substrate-proximal poles of the base cells (the bright dots at the base of the microcolony) (Fig. 4b) in this case. Moreover, the default of the program discards short fibers below a preset threshold. Such features of the program left the bright dots not effectively captured on the ctFIRE output images.

**Time-lapse video analysis.** The custom image analysis workflow for objectively batch processing the time-lapse videos was developed in JEX[46,47], an open-source image analysis software that uses well-established libraries from ImageJ. Workflows are available upon request. Briefly, raw masks for identifying microcolonies were generated in two ways to robustly accommodate variation in bright-field imaging. In the first approach, bright-field images were gamma adjusted (γ = 0.7), inverted, background subtracted, Gaussian-mean filtered, and thresholded. In the second, images were background subtracted, inverted, Gaussian-mean filtered, and thresholded. The two masks were then combined using an OR operation to form the final mask. Microcolony objects were then tracked in JEX using the fast LAP tracking algorithm of TrackMate[48]. JEX was then used to calculate the masks and Euclidean distances needed for calculating edge-to-edge displacement (Supplementary Fig. 2). The center-of-mass locations, areas, and edge-to-edge displacements of each microcolony were then quantified. Only microcolonies within a circular region of interest (ROI) (1.84 mm in diameter) defining the inner region of the microdrop (2 mm in diameter) were quantified to avoid image artifacts associated with the edge of the microdrop. Raw data was then exported to.csv for analysis in R/RStudio. The equivalent diameter of each microcolony was calculated from the microcolony area ($D$ = 2sqrt(area/π)). Equivalent microcolony volume was calculated using the equation for a sphere ($V = (4/3)\pi(D/2)^3$). Equivalent volume was used as the weight for calculating volume-weighted means where necessary. For visualization, plots over time were smoothed using the 'smooth.spline' function of the R 'stats' package with a smoothing parameter spar = 0.4. Microcolonies that were below 10 µm were filtered from the data prior to plotting and summarizing because the average size of a single bacterium was 6.7 µm. Log-scaling was chosen given that the variability/noise in the data (versus measurement uncertainty) scaled with the signal.

**Quantification of base cells.** Bright-field images (40× objective with 1.5× tube lens) of microcolonies from the hyper-motile mutant fjoh_0352 were recorded on microscope (Nikon Eclipse Ti) at 12 h after starting culture, with the focal plane set at the base of the microcolonies on the surface. Note that confocal microscopy was not adopted to capture the 3D orientation of base cells because the stroking movement of the base cells (at about 2 Hz) is way faster than the scanning speed of the microscope. Image analysis was executed in Fiji ImageJ. First, the background was subtracted with rolling ball radius = 50 pixels and a light background. Then the contrast of the image was enhanced by saturated pixels = 3.0%. Two circular ROIs were set to pick the microcolony diameter ($D$) and base diameter ($d$), respectively (Fig. 4b, Supplementary Fig. 5a). The area of the larger ROI was measured and then converted to the equivalent diameter ($D$ = 2sqrt(area/π)). The bright dots (i.e., base cells) within the smaller ROI were picked using "Process → Find Maxima". The prominence was adjusted on each image until the picking followed the distribution of the bright dots. Plotting of the number of base cells ($N$) as a function of microcolony diameter ($D$) and power fitting was done in Excel to show the power-law dependence between $N$ and $D$ (Fig. 4c). The intuitive hypothesis (Hypothesis A) upon the relationship between the number of base cells ($N$) and the microcolony diameter ($D$) is $N \propto D^2$, in which the surface density of the base cells ($n$, unit: cells per µm$^2$) is assumed constant. Another potential relationship (Hypothesis B) between $N$ and $D$ is $N \propto D$. We resolved $n = 0.25$ using the data point of $D = 10$ µm, which translates to a single base cell per 4 µm$^2$ of microcolony surface area. For Hypothesis A, $N = n\pi[(Avg. d/D)D/2]^2 = 0.05D^2$, where Avg. $d/D = 0.48$ is the averaged ratio between $d$ (diameter of the base of a microcolony) and $D$ measured in experiment (fjoh_0352 microcolonies at 12 h after initiating culture, Supplementary Fig. 5b, c). For Hypothesis B, $N = nD = 0.25D$. In this case based on Stokes law (i.e., $F_{Drag} = 6\pi\mu Rv$, where $F_{Drag}$ is drag force, $\mu$ is dynamic viscosity of the fluid, $R$ is the radius of the moving particle, $v$ the particle velocity), the microcolony velocity should remain constant and independent of the size change of the microcolonies (because $v \propto F_{Drag}/R \propto N/D$ = Const.), which would conflict with data shown in Fig. 2f. We analyzed and plotted the experimentally measured $N$ as a function of $D$ (Fig. 4c) and applied power fitting to get the power-law dependence between $N$ and $D$. The line on the plot shows the power fitting as $N = 0.2D^{1.36}$, which not only supports the increasing velocity with increasing microcolony size in Fig. 2f but also is an important input for the computer simulation. The simulated results between microcolony velocity and microcolony size (Supplementary Fig. 7a) showed a consistent trend compared with the experimental results shown in Fig. 2f.

**Effect of CCCP quantification.** Single frame bright-field images of microcolonies at 12 h and 15 h from both WT and fjoh_0352 were obtained from time-lapse movies of microdrops in the presence or absence of CCCP. Microcolony size and abundance were quantified in Fiji ImageJ. First, the image was converted to 8-bit and a threshold applied. Second, the image was analyzed using the function "Analyze → Analyze Particles…" Finally, ROIs were manually curated to remove

the errantly placed ones and to include microcolonies that were missed by the software. The data was exported and only microcolonies greater than 1000 pixels[2] were included in the analysis to only quantify the moving and merging of mature microcolonies and not the formation of new microcolonies. Conversion of pixels to μm and translation into diameter ($D$) using the area of a circle equation gives a mature microcolony baseline of $D > 37$ μm. The data were graphically represented and descriptive statistics were acquired in R/RStudio using the packages *ggplot2* and *psych*, respectively.

**Governing equations in the computer simulation**. The computational domain (i.e., the under-oil microdrop with bacteria or microcolonies) is represented as $\Omega = \Omega_f \cup \Gamma$, where $\Omega_f$ is the sub-domain occupied by the fluid medium, and $\Gamma$ denotes the boundaries of the domain found between the medium and a secondary phase (e.g., oil, bacteria, or microcolonies). Consider the domain $\Omega$ contains $N_s$ bacteria (or microcolonies), with each of them bearing a separate boundary $\Gamma_n, n = 1, \dots, N_s$. Assuming finite separation between any two bacteria (or microcolonies), the boundary of the domain can be partitioned into a disjoint union $\Gamma = \Gamma_m \cup \Gamma_1 \cdots \cup \Gamma_{N_s}$, where $\Gamma_m$ denotes the boundary of the underoil microdrop between the medium and oil. Each bacterium (or microcolony) has a position $X_n$, and undergoes rigid-body kinematics prescribed by a translational velocity $V_n$. Due to the low Reynolds number (Re $\approx 1e-6$), it can be posed as a Stokes problem. Thus, the motion of bacteria (or microcolonies) and hydrodynamic coupling with the fluid medium are governed by the steady, boundary-value Stokes equation:

$$
\begin{aligned}
\frac{\nabla p}{\rho} - \nu \nabla^2 u &= 0, \ for\, x \in \Omega_f \\
\nabla \cdot u &= 0, \ for\, x \in \Omega_f \\
u &= 0, \ for\, x \in \Gamma_m \\
u &= V_n, \ for\, x \in \Gamma_n, n = 1, \dots, N_s
\end{aligned} \tag{1}
$$

subject to the force-free and torque-free constraint on each bacterium (or microcolony) as:

$$
\begin{aligned}
\int_{\Gamma n} \sigma \cdot dA &= 0 \\
\int_{\Gamma n} (x - X_n) \times (\sigma \cdot dA) &= 0
\end{aligned} \tag{2}
$$

here, $u$ and $p$ are the fluid velocity and pressure, respectively; $\nu$ is the kinematic viscosity of fluid medium; $\rho$ is the density of the fluid medium; and $\sigma = -pI + \nu[\nabla u + (\nabla u)^T]$ is the total stress exerted by fluid medium on each bacterium (or microcolony). By solving Eqs. (1) and (2) concurrently, we can obtain the translational velocity $V_n$ of each bacterium (or microcolony) as well as the fluid velocity $u$ and pressure $p$.

**Numerical method**. To numerically solve Eqs. (1) and (2), the generalized moving least squares (GMLS) discretization method was employed. In that, the fluid domain $\Omega_f$ is discretized by a set of collocation points (called GMLS nodes). Another set of GMLS nodes are distributed along the boundaries $\Gamma = \Gamma_m \cup \Gamma_1 \cdots \cup \Gamma_{N_s}$ for imposing the boundary conditions and constraints in Eqs. (1) and (2). In the GMLS discretization, for a GMLS node at $x_i$ and a given function $\psi$ evaluated at its neighbor locations: $\psi_j = \psi(x_j)$, a polynomial $\psi_h(x)$ of order $r$ is sought locally to approximate $\psi$ and its derivatives $D^\alpha \psi$ at $x_i$. To this end, $\psi_h(x) = P^T(x)c^*$ with a polynomial basis $P(x)$ and coefficient vector $c^*$ such that the following weighted residual functional is minimized:

$$
J(x_i) = \sum_{j \in N_i} [\psi_j - P_i^T(x_j)c_i]^2 W_{ij} \tag{3}
$$

As such, for $\psi \in span(P(x))$, $\psi$ can be exactly reconstructed. From this polynomial reconstruction property, it follows that high-order accuracy can be achieved via the GMLS approximation by taking large $r$, e.g., $r = 4$[49]. Following standard arguments for the minimization of a symmetric positive definite quadratic form, the solution of Eq. (3) is given by:

$$
\psi_h(x) = P_i^T(x)c_i^*
$$

with

$$
c_i^* = \left( \sum_{k \in N_i} P_i(x_k) W_{ik} P_i^T(x_k) \right)^{-1} \left( \sum_{j \in N_i} P_i(x_j) W_{ij} \psi_j \right) \tag{4}
$$

To approximate derivatives of the underlying function, an arbitrary $\alpha$th order differential operator can be written as[50]:

$$
D^\alpha \psi(x) \approx D_h^\alpha \psi_h(x) := (D^\alpha P(x))^T c^* \tag{5}
$$

In Eqs. (3) and (4), the weight function $W_{ij} = W(r_{ij})$, where $r_{ij} = \|x_i - x_j\|$, and $W(r) = 1 - (\frac{r}{\epsilon})^4$ for $r < \epsilon$ with $\epsilon$ the compact support ($W(r) \equiv 0$, for $r \geq \epsilon$). Hence, it is only necessary to include the GMLS nodes within an $\epsilon$-neighborhood of the $i$th GMLS node, i.e., $j \in N_i = \{x_j s.t. r_{ij} < \epsilon_i\}$. By choosing appropriate polynomial basis and requiring the approximation to be exact for polynomials up to a given order, e.g., $r = 4$, GMLS can achieve arbitrarily high-order accuracy through a weighted least square optimization. To enforce compatibility with the

incompressible fluid constraint, the fluid velocity $u$ is directly constructed from a divergence-free polynomial basis. The pressure $p$ in the Stokes equation is approximated via a staggered discretization analogous to finite volume methods. Adaptive refinement in spatial discretization draws on *a posteriori* error estimator as the adaptive criterion and a four-step adaptive algorithm: SOLVE → ESTIMATE → MARK → REFINE, similarly to that in adaptive finite element methods. More details about the numerical method can be found in literature[27,28] and the authors' previous work[26].

**Validation of the numerical method**. A benchmark test was conducted to validate and demonstrate the accuracy of the employed numerical method. In particular, we examined Wannier flow[51], a Stokes flow confined between two rotating cylinders. This specific test was chosen to validate the numerical method because its analytical solution is available[51]. The radii of the two cylinders are $r_1$ and $r_2$, respectively; the two cylinders are non-concentric with $\Delta s$ the distance between their centers; the minimum gap width between the two cylinders' surfaces is hence given as $r_2 - r_1 - \Delta s$; and, the two cylinders are rotating in the same direction with different constant rotational speeds (Supplementary Fig. 6a). By adjusting the position of the inner cylinder, the minimum gap width between the two cylinders' surfaces varies. We computed the drag force exerted on the inner cylinder by the fluid at varying gap widths. The numerical predictions show good agreement with the analytical solution[51] for all the cases considered (Supplementary Fig. 6b). The minimum gap width between the two cylinders examined herein can be as small as $r_1/20$, for which many numerical solutions could fail due to the singularity arising from the narrow gap. The high-order accuracy and spatially adaptive refinement capability of the employed numerical method ensured the accurate solution for such a challenging case. For this validation and all simulations in this work, the 4th-order ($r = 4$) GMLS discretization was employed for solving the Stokes equation, which ensured the 4th-order accuracy and convergence for the numerical solutions.

**Computational simulation**. The equivalent level of base cells in microcolony motility that participate in force generation was simulated using the spatially adaptive generalized moving least squares (GMLS) method (Fig. 4d, Supplementary Fig. 7c), which has been demonstrated to provide high-order accurate and stable solutions for suspension flows at low Reynolds number (Re $\approx 1e-6$). Details for the Stokes equations, numerical method, and validation of the simulations can be found in SI. Considering the bacteria and microcolonies move along a plane surface, the simulation was conducted based on a two-dimensional model. Microcolony velocity in the culture time window (11.5–12.5 h) (Supplementary Fig. 7a), the number of base cells ($N$) as a function of microcolony diameter ($D$) (Fig. 4c), microcolony density for a given $D$ (Supplementary Fig. 7b), and the size, density, and gliding velocity of single cells (Supplementary Fig. 1), all experimentally measured from the hyper-motile mutant *fjoh_0352*, were the input data. The net propulsion force on a microcolony ($F_{Prop,microcolony}$) and its theoretical maximum (Max. $F_{Prop,microcolony}$) were calculated from the collective movement of microcolonies and the gliding of individual bacteria, respectively, and with NFP defined as $F_{Prop,microcolony}/$Max. $F_{Prop,microcolony} \times 100\%$.

**Reporting summary**. Further information on research design is available in the Nature Research Reporting Summary linked to this article.

## Data availability

All data that supported these findings are all available from the corresponding authors upon reasonable request. Specifically, the time-lapse videos are 5 min intervals for 24 h and, with the subsequent analysis, together yield TB of data. Representative videos can be found in the supplemental material and the rest can be made available upon request. Source data are provided with this paper.

## Code availability

The softwares used for data acquisition, data analysis, and computational simulation are available upon request.

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

## Acknowledgements

This work was supported by NSF EFRI-1136903-EFRI-MKS, NIH R01 EB010039 BRG, NIH R01 CA185251, NIH R01 CA186134, NIH R01 CA181648, EPA H-MAP 83573701, and American Cancer Society IRG-15-213-51 (Beebe Lab), and NSF MCB-1243671, U. S. Army Research Laboratory and the U. S. Army Research Office under contract/grant number W911NF1910269, and USDA NIFA Postdoctoral Fellowship grant no. 2019-2018-08058/accession no. 1019190 (Handelsman Lab). We are grateful to Dr. Mark McBride and his lab at the University of Wisconsin-Milwaukee for providing the *sprB* and *remA* mutants. We thank Dr. Manuel Garavito (Handelsman Lab) for assistance in preparing the bacterial strains, and Mr. Zachary Hite (Beebe Lab) for assistance in preparing the UOMS devices and microscopy.

## Author contributions

C.L. discovered the social motility in the gliding bacterium *Flavobacterium johnsoniae*. C.L., A.H., and G.L.L. designed the mutant analysis and identified the gliding apparatus that regulate microcolony formation and motility. G.L.L. constructed the *gldD* and *fjoh_0352* mutants. A.H. cultured and processed all the bacterial strains. C.L. prepared the UOMS devices and collected the data on growth dynamics. C.L. and J.M.A. performed multiphoton microscopy and identified the 3D structure of microcolonies. C.L. and A.H. performed lectin staining and identified the biofilm. A.H. conducted image analysis to quantify the effect of CCCP on microcolonies and performed autoaggregation assays. C.L. collected the data on base cells and identified the orientation and movement of base cells. C.L. collected the data on the single-cell organization of *F. johnsoniae*. C.L. and J.W.W. performed single-cell organization analysis and data visualization, and quantified single-cell parameters (i.e., cell length, width, straightness, and alignment). C.L. and J.W.W. designed the particle tracking algorithm. J.W.W. performed batch processing of the time-lapse videos and data visualization, and quantified growth dynamics and microcolony motility. W.H., W.P., and C.L. designed the computational studies. W.H. established the numerical models and performed the simulations. D.J.B. and J.H. supervised experimental design, data analysis, and data presentation. All authors wrote and revised the manuscript.

## Competing interests

D.J.B., J.H. and A.H. have potential competing interests related to technologies presented here. D.J.B. holds equity in BellBrook Labs LLC, Tasso Inc., Stacks to the Future LLC, Lynx Biosciences LLC, Onexio Biosystems LLC, Turba LLC, Flambeau Diagnostics LLC, and Salus Discovery LLC. D.J.B. is a consultant for Abbott Laboratories. J.H. and A.H. hold equity in Wacasa, Inc. J.H. is a consultant for Phylagen, Inc. The remaining authors declare no competing interests.

**Additional information**

