## [Peer Review File · Nature Communications]

Reviewers' comments:

Reviewer #1 (Remarks to the Author):

I'm not sure what to say about this paper. I don't like the title, at all, but I don't have a good alternative. I don't like the name *zorb*, but if that is what the authors want to call it, I guess it doesn't make much of a difference. I can't tell whether or not the effect is an artifact of the set-up, but regardless, the result is interesting. The system and resulting biology seems robust, but I don't know how generalizable/relevant this is, and I'm really not sure what else one could do with it or do to follow it up experimentally. So ultimately I think this is just a parlor trick, but even if so, it is still a darn cool trick and it advertises *F. johnsoniae* which is a great organism that deserves more attention.

Besides vague gut-level discomfort with the presentation, biology and relevance, I have a problem with the mechanism. I can think of three ways force could be generated and the walking "power stroke" proposed (where force is coming from the aggregate-proximal pole) seems the least likely to me. I'm just not sure the rotational force of tethered individual cells one pole (anchored in the colony) generates sufficient translational torque at the other pole (bound to the substrate) to propel a large mass. Another mechanism is that cells bind to the substrate by their pole and movement of focal adhesion complexes through the bound pole drives translocation of just the pole (where force is coming from the substrate-proximal pole). So it is like a cell is laying longitudinally on the surface but the longitudinal surface is reduced to just a pole. While cells normally rotate under conditions of being bound like this, anchoring of the other pole in the mass may allow translocation instead. Still in this case, I could see the pole moving on the surface but I don't see how the force would be rigidly transmitted to the mass. In either mechanism of active force generation, I'm not sure how directional net translocation is achieved.

The most likely scenario instead, I think, is passive processive adhesion (on which gliding motility depends) and de-adhesion, driven by Brownian motion of the mass, that has nothing to do with gliding motility per se. How responsive is the colony movement to temperature changes?

The difference in all mechanistic models I can think of relies on distinguishing the contribution of focal adhesion and focal adhesion complex translocation (gliding). To do this one would need a motor-deficient mutant that both adheres and coheres, but does not glide. I believe the current manuscript uses a *gldD*, *sprB* and *remA* mutants which have defects in adhesion. It seems certain alleles of *gldJ* have the property of adhesion but not gliding and might be tested. Without a "motor" mutant, it is hard for me to believe that the biofilm is actually "walking" in the way the authors claim.

Minor comments:

Where did *fjoh_0352* come from? The mutant is presented in the section of mutants defective in gliding motility but instead has hyper-aggregate movement so I'm not sure why this mutant was tested or how it was found. Nice sensitized background for future study though.

Consider adding myxobacteria to the discussion as they are the classical case of a mobile biofilm.

Reviewer #2 (Remarks to the Author):

The manuscript by Li et al describes a novel multicellular motility behavior of the gliding bacterium *Flavobacterium johnsoniae*. This involves aggregation of the rod-shaped cells into large aggregates (zorbs). The 'pinwheeling' motion of individual rod cells of the aggregate that contact the substratum propel the aggregate. A recently developed microscopy technique facilitated discovery of the phenomenon. The results are in general convincing, including the movements of the aggregates ('biofilms', 'zorbs') being propelled by the motions of individual cells near the base of

the aggregate. The enhanced zorb movement of the Fjoh_0352 mutant was particularly impressive. The resulting 'walking biofilms' may have important implications. Concerns and suggestions follow.

1) The authors should indicate more clearly the nature of the mutations in the newly constructed *gldD* and *Fjoh_0352* mutants. These were obtained via a transposon mutagenesis screen, but the nature of the mutations (insertion near beginning or end of gene for example?) and the effect if any of the transposons on surrounding genes is never mentioned. Could these have been polar mutations that exerted their effects by disrupting expression of downstream genes for example? At minimum the site of insertion within each gene, and gene context (potential operon structure) should be described somewhere, perhaps in the supplemental materials.

2) The Shrivastava et al 2012 paper (J. Bacteriol. 194:3678-3688) referenced in the Supplemental section should have been cited in the main text, and discussed in relation to the results observed in the current manuscript (cell aggregate formation and roles of *SprB*, *RemA*, and polysaccharides). Each of these were mentioned in the Shrivastava 2012 paper, using some of the same mutants that the authors used. For example, in the 2012 paper, Shrivastava et al also demonstrated a link of aggregate formation to cell-surface polysaccharides, since mutations that disrupted specific polysaccharide synthesis or secretion genes (*remC*, *wza*, *wzc*) prevented aggregate formation. These points seem relevant to this work on aggregate (zorb) formation and movement of these aggregates, and the roles of *RemA*, *SprB*, and of *fjoh_0352* (*wzz* homolog). The results are not duplicative, but they are probably related, and should thus be briefly discussed.

3) "previously reported in *F. johnsoniae* (i.e., spinning along an axis with a tethered end)¹⁴"

Reference 14 describes this behavior but is not the best reference if only one reference is used, for two reasons. First, Ref 14 used 'Cytophaga' strain U67. This is probably a member of the genus *Flavobacterium*, but is probably not *F. johnsoniae*. Also, a paper on *Flavobacterium johnsoniae* (then called 'Cytophaga johnsonae') published 3 years earlier also described this phenomenon. This reference (which corresponds to your current ref 16) is:

Pate JL and LYE Chang. 1979. Evidence that gliding motility in prokaryotic cells is driven by rotary assemblies in the cell envelopes. *Curr. Microbiol.* 2:59-64.

You could use the Pate reference here, or use both references (Pate and Lapidus).

4) "By contrast, the majority of cells within the zorbs align with random orientation, except for the cells at the base of the zorb, which align perpendicular to the surface."

Clearly, all of the cells at the base do not align perpendicular to the surface. They have many different orientations with respect to that surface; they do not all form a 90 degree angle with respect to that surface. In fact, none of them form a permanent 90 degree angle with respect to the surface (or the zorbs would not be able to move much).

Suggest change to something like:

"By contrast, the majority of cells within the zorbs align with random orientation, except for some at the base of the zorb, which align perpendicular to the surface."

5) Second sentence of discussion, suggest change:

"non-piliated bacterial species"

to

"non-piliated bacterium"

6) Near end of discussion suggest change:

"thought to aid *C. canimorsus* traverse host tissue"

to something like

"thought to help *C. canimorsus* traverse host tissues"

7) Supplemental Fig 3 title

"Growth dynamics and motility analysis of sprB and remA zorbs."

It will not be clear to everyone that 'sprB and remA zorbs' refers to 'zorbs of sprB and remA mutants'.

I suggest changing to either

"Growth dynamics and motility analysis of zorbs of sprB and remA mutants."

Or

"Growth dynamics and motility analysis of sprB and remA mutant zorbs."

Reviewer #3 (Remarks to the Author):

The manuscript by Li et al reports on the novel type of collective bacterial motility. Right away we would like to state that although the observations are absolutely exciting, the level of scientific inquiry into the origins of the observed mechanism are not satisfactory. In our view, the authors stopped halfway in their effort to understand the phenomenon at hand. The proposed mechanism of force generation and propulsion of microcolonies remains largely at the level of hypothesis. We provide our detailed comments in chronological order below.

1. In the abstract, formulation "...indicates ... by walking on coordinated ... cells" although carefully worded is still an overstatement. The arguments provided by the authors in the manuscript do not necessitate coordination as the only possible explanation of the effect, see point 10 for details.

2. In the abstract, last sentence mentioning implications for host pathogen interactions and antibiotic resistance is also too far fetched. First, there is still many things to clarify about the mechanism and second, there is no concrete link to neither of those.

3. In Fig 1 b, only 3 slices of the z-stack are shown. It would be desired to see full 3D rendering of the images to confirm if the exact shape of the aggregates really matches the sketches by the authors.

4. Authors use ConA labelling to claim the presence of the EPS. To us it does not seem as a sufficient proof that there is EPS present, or how developed it is. In the same context, in the movies, when observing the merging of aggregates one sees their rapid rounding up. Would this fast relaxation to the circular (spherical) shape consistent with the presence of the EPS which is supposed to provide extra mechanical stability to the aggregate? What happens to the EPS during dissolution? What triggers the dissolution process?

Overall, the presence of the EPS is only mentioned in passing in this study, while it certainly deserves more attention. We note, however, that it might be out of scope of this paper.

5. Line 65. Some advanced image analysis is mentioned in the text, in particular about merging and splitting, growing and shrinking. But in fact no data or analysis on those events is presented - neither in the text, nor in SI.

6. We would like to question if the spider plots and velocity of aggregates are the right (or sufficient) quantities to quantify their motility. The motility of aggregates resembles (active) random motion. This is typically characterised by the mean squared displacement (and other quantities). As the Fig 2 f clearly shows, the velocity of the aggregate is itself a broadly distributed random variable. It would be instructive to see the probability distribution function of those.

Authors mention persistence in the text, but this was never quantified. The typical quantities to look in this case are the directional and velocity autocorrelation functions. Thus overall for this point, the statistical analysis of the aggregates' motility should be significantly expanded.

7. Fig. 2. What is the reason to use the log scale in d and f if the quantities are changing only about one order of magnitude?

8. Line 106: In Fig. a and b, it is very hard to see the circles mentioned by the authors as the "standing" cells. Moreover, that statement seemingly contradicts images in SI, fig 1, which show shorter but still flat lying cells. The statement about "legs" seems to be central for this manuscript. Thus it deserves a better experimental quantification. With a given size of the colonies, confocal or TIRF imaging of the near surface layer of cells at single cell resolution should be feasible. That would clearly demonstrate the fraction and alignment of the cells at the surface.

9. Fig. 3b. What is the rationale for seeing a particular scaling of the number of legs with zorb diameter?

10. One of the points where we would like to strongly disagree and that might be very misleading is about "highly-organised", "coordinated movement of legs" that leads to achieve "directional persistence". First, this persistence is not quantified in the manuscript and most probably is based on the visual impression from the movies. Second, and probably most important, an apparent persistence in the movement of aggregates does not imply any "coordination". As was demonstrated in the context of *Neisseria* aggregates driven by cycles of elongation and retraction of pili filaments, force balance at the level of both, single cells and microcolonies of many cells, may lead to persistent motion without any coordination between stochastic individual pili forces (Pönisch et al 2019). Therefore we believe that the statements of the last paragraph before discussions are misleading and need serious rethinking/reformulation/tests.

11. Concerning the modeling, the authors refer to Ref 18, which, even if a reader tries to go through it, won't really help to understand how the simulations are implemented with respect to this concrete problem of zorbs. Unfortunately there are no further details on this method in the manuscript and thus we can not accept the results of simulations to bare any predictive power.

13. While the pinwheel motion of tethered cells sounds like a plausible mechanism of force generation, most of the aspects of how that microscopic force generation results in the macroscopic motion of the colonies is not really explained, and we dare to say not understood. How individual force-dipoles tethered to the zorb propel the whole zorb is not explained or modeled. If there is a coordination between individual "legs" how is that achieved? When they make a stroke - are they synchronised? are they spatially correlated? All those, in principle, are central to understanding the mechanism and can be experimentally accessed and quantified.

14. On the movies of aggregation process it is easy to notice events of aggregates being attracted to each other before they even touch. What is the mechanism of that (and certainly that is also related to the motility mechanism). What are the forces driving the merging of aggregates?

15. In the Discussion, concerning the movement of colonies, *P. earuginosa* is another well known example (Sauer et al 2002).

To conclude, we feel that many of the above points could be addressed during the revision, while the most challenging question remains - what is the actual mechanism of how this type of collective motility works. To answer that, a significant amount of further experimental and theoretical work needs to be invested (points 10-14 in particular). Thus, at this stage, we can not recommend this paper for publication in Nature Communications.

To All Reviewers:

We thank the reviewers for their time and effort in reviewing the original manuscript and appreciate their constructive and overall quite positive comments. As recognized by all reviewers, the biofilm-like multicellular structure represents a novel type of collective movement in a gliding bacterium. This work is relevant for the study of prokaryotic motility, biofilm dynamics, bacterial cargo transport, microbial adaptation, and other related areas.

In addition to their positive comments, the reviewers expressed some concerns regarding the clarity of presentation, quantification of the physics, and impact of the work. In summary, we added a new figure to the main text and three main text figures were amended to improve clarity. In addition, we added three figures to SI to further address reviewer concerns. We are incredibly grateful for these comments as they have driven a major restructuring of the main text leading to a manuscript that we believe is greatly improved. Below we provide a point-by-point response to the reviewers' comments. **Response to reviewer comments shown in "red".**

Reviewer #1

I'm not sure what to say about this paper. I don't like the title, at all, but I don't have a good alternative. I don't like the name zorb, but if that is what the authors want to call it, I guess it doesn't make much of a difference.

Thank you for the feedback. We've updated the title to "Social biofilm motility in a gliding bacterium" and have changed the terminology from "zorb" to the more traditional "microcolony."

I can't tell whether or not the effect is an artifact of the set-up, but regardless, the result is interesting.

We agree that the signals regulating microcolony development are unknown and have addressed the reviewer's concerns with the following sentence in the discussion: "While microcolony development could be caused by stresses generated by the under-oil microenvironment, and thus could be an environment-dependent adaptation, the discovery of motile microcolonies generated by a gliding bacterium expands our understanding of prokaryotic collective behavior."

The system and resulting biology seems robust, but I don't know how generalizable/relevant this is, and I'm really not sure what else one could do with it or do to follow it up experimentally. So ultimately I think this is just a parlor trick, but even if so, it is still a darn cool trick and it advertises *F. johnsoniae* which is a great organism that deserves more attention.

We are glad the reviewer finds the microcolonies cool - so do we! To address concerns about the generalizability of microcolonies, we have removed language suggesting other gliding bacteria could form microcolonies to focus on the relevance for *F. johnsoniae*. Specifically, we include the following sentence in the discussion, "*F. johnsoniae* is a common rhizosphere resident and whether this collective motility occurs on plant roots or soil particles, the native environment for *F. johnsoniae*, remains to be observed." Furthermore, we have data for a second paper that displays interactions between *F. johnsoniae* microcolonies and rhizosphere co-resident *Bacillus cereus*. The emergent property is quite striking and, while it's too much information for this paper, suggests ecological relevance for microcolonies.

Besides vague gut-level discomfort with the presentation, biology and relevance, I have a problem with the mechanism. I can think of three ways force could be generated and the walking “power stroke” proposed (where force is coming from the aggregate-proximal pole) seems the least likely to me. I’m just not sure the rotational force of tethered individual cells one pole (anchored in the colony) generates sufficient translational torque at the other pole (bound to the substrate) to propel a large mass. Another mechanism is that cells bind to the substrate by their pole and movement of focal adhesion complexes through the bound pole drives translocation of just the pole (where force is coming from the substrate-proximal pole). So it is like a cell is laying longitudinally on the surface but the longitudinal surface is reduced to just a pole. While cells normally rotate under conditions of being bound like this, anchoring of the other pole in the mass may allow translocation instead. Still in this case, I could see the pole moving on the surface but I don’t see how the force would be rigidly transmitted to the mass. In either mechanism of active force generation, I’m not sure how directional net translocation is achieved.

We thank the reviewer for these comments; they were a driving force in the restructuring and re-writing of our manuscript. We agree that, at this time, these other mechanisms can not explicitly be ruled out. Comparing our proposed mechanism to possible alternatives, we believe strengthens our arguments and illuminates next steps for this project. We have changed language throughout the results section and included the following paragraph in the discussion, “How the activity of gliding apparatus from individual *F. johnsoniae* cells translocates to the movement of an entire microcolony remains unknown. The movement of the microcolonies via base cells briefly gliding when attached to the surface has not been conclusively excluded. Additionally, the many base cells may act as focal adhesions successively binding to the substrate, not unlike the pili used by *Neisseria*. However, due to the observations of pinwheeling and the dynamic nature of the cells within the microcolony, we propose the hypothesis of a “power stroke” force originating from base cells tethered to the microcolony (not surface) by their rotary gliding motors that propel the microcolony forward upon contact with the surface. Regardless of mechanism, the computational simulations demonstrate the extent to which intercellular synchronization is required among the base cells that leads to translocation of the microcolony on a millimeter scale. Future experiments with fluorescent WT cells to track behavior of single cells within the microcolony, especially at the base, will begin to differentiate among the possible mechanisms.”

The most likely scenario instead, I think, is passive processive adhesion (on which gliding motility depends) and de-adhesion, driven by Brownian motion of the mass, that has nothing to do with gliding motility per se. How responsive is the colony movement to temperature changes?

Again, we are grateful for the opportunity to clarify our arguments based on this reviewer’s feedback. We explicitly describe how Brownian motion is likely not a driving force in microcolony motility in the results, “Compared to the active mechanism proposed where the single-cell movements in the microcolonies fuel microcolony movement, it’s necessary to clarify the probability of whether the collective movement could be driven by Brownian motion, a common passive mechanism of particle diffusion. The mean squared displacement of particles in Brownian motion is inversely proportional to the particle size, but in the current system, the velocity correlates positively with microcolony size (Fig. 2f). In addition, the microcolony velocity decreases in the late stage (> 15 hr) with a diminishing microcolony size due to the dispersal. Therefore, Brownian motion is not a dominant mechanism in the microcolony movement.”

To further exclude Brownian motion, we added Fig. 3 entitled “**Proton motive force (PMF) uncoupler shows microcolony motility and merging is an active process,**” which shows WT microcolony motility and ability to merge depends on PMF by quantifying the effects of a PMF-inhibitor (CCCP) on microcolony size and number over time. We describe the results in a new section called “**Energy source of mature microcolony motility**”. We show “mature microcolonies quantified at 12 hr and 15 hr display an increase in size concurrent with a reduction of total number over time, demonstrating the ability of the microcolonies to migrate towards one another and merge to form larger, but fewer, microcolonies” and that in the presence of the PMF-inhibitor “the WT microcolony growth is abrogated (Fig. 3b column 3,4, Supplementary Movie 5, Supplementary Movie 6) whereas the overall abundance of microcolonies is increased (Fig. 3b column 2,4), suggesting normal merging and motility requires the proton motive force and, thus, gliding. While not completely abolished, the *fjoh_0352* microcolony growth from 12 hr to 15 hr is reduced in the presence of CCCP (Fig. 3b column 7,8, Supplementary Movie 7, Supplementary Movie 8) compared to growth in its absence (Fig. 3b column 5,6) and the overall abundance of microcolonies is increased (Fig. 3b column 6,8), similar to WT.”

The difference in all mechanistic models I can think of relies of distinguishing the contribution of focal adhesion and focal adhesion complex translocation (gliding). To do this one would need a motor-deficient mutant that both adheres and coheres, but does not glide. I believe the current manuscript uses a *gldD*, *sprB* and *remA* mutants which have defects in adhesion. It seems certain alleles of *gldJ* have the property of adhesion but not gliding and might be tested. Without a “motor” mutant, it is hard for me to believe that the biofilm is actually “walking” in the way the authors claim.

We agree with the reviewer that a gliding motor-deficient would demonstrate the reliance of microcolony motility on active motor function. To address this, we have used a biochemical approach described above and shown in our new Fig. 3.

Where did *fjoh_0352* come from? The mutant is presented in the section of mutants defective in gliding motility but instead has hyper-aggregate movement so I’m not sure why this mutants was tested or how it was found. Nice sensitized background for future study though.

We have improved the clarity of mutant selection in our results section and conducted a physiological assay (autoaggregation) to further categorize *fjoh_0352*. The following paragraph was added to the results section, “Since RemA has been suggested to mediate cell-cell interactions and demonstrates defective microcolony motility, we explored the impact of other physical disruptions in *F. johnsoniae* collective motility. From a collection of *F. johnsoniae* transposon mutants, we selected *fjoh_0352*, which contains a mutation in a homolog of *wzz* and is predicted to construct the O-antigen of lipopolysaccharides (LPS). O-antigen mutants in *M. xanthus* reduce social motility¹⁸ but, to our surprise, disruption of *fjoh_0352* had the opposite effect in *F. johnsoniae*. Mature *fjoh_0352* microcolonies exhibited a dramatic hyper-merging (Fig. 2b-d), hyper-motile (Fig. 2e-f) phenotype resulting in massive microcolony formation (Fig. 2d solid lines, Supplementary Movie 4). We suggest that the absence of polar oligosaccharides and exposure of lipid A of LPS mediates increased hydrophobicity of cells in the *fjoh_3052* mutant, which in turn mediates stronger cell-cell interactions or binding within, and between, microcolonies. Furthermore, we found *fjoh_0352* exhibits greater autoaggregation compared to both WT and *gldD*, which could be the result of enhanced cell-cell interactions (Extended Data Fig. 4). Increased cell-cell interactions were similarly observed in *M. xanthus* O-antigen mutants¹⁸. Further genetic dissection and uncoupling of LPS,

secreted polysaccharide pathways, and gliding will contribute to developing a model for cell-cell interactions within the motile microcolonies.”

Consider adding myxobacteria to the discussion as they are the classical case of a mobile biofilm.

We agree with the reviewer that *Myxobacteria* are excellent models for both gliding and social motility. We’ve included the following sentences in our introduction to frame the novelty of our motility by overviewing the current models, including *Myxococcus*, after our second introductory sentence explaining biofilms are typically stationary while individual cells can translocate over a surface. “In some cases, these seemingly disparate behavioral states intersect. Small *Pseudomonas aeruginosa* microcolonies are dynamic, using type-IV pili to merge and split before settling down to form a sessile biofilm⁹. *Myxococcus xanthus* displays diverse multicellular and motile behaviors. Individual cells rely on gliding motility whereas social motility for bacterial predation and the formation of fruiting bodies primarily depends upon type-IV pili⁶. *Neisseria* species also use type-IV pili to form large microcolonies that move across a surface as discrete units and grow via merging^{10,11}. In these examples, type-IV pili are important contributors to the movement of microcolonies.” Additionally, we draw parallels between *M. xanthus* with *F. johnsoniae* in the LPS results section (see answer to previous comment). Finally, we include comparisons to *Myxococcus* in our discussion to contextualize our microcolony results, “Collective motility of gliding bacteria has been studied most thoroughly in *Myxococcus*, which coordinates rotary gliding motors and type-IV pili to form predatory swarms and fruiting bodies.”

Reviewer #2

- 1) The authors should indicate more clearly the nature of the mutations in the newly constructed *gldD* and *Fjoh_0352* mutants. These were obtained via a transposon mutagenesis screen, but the nature of the mutations (insertion near beginning or end of gene for example?) and the effect if any of the transposons on surrounding genes is never mentioned. Could these have been polar mutations that exerted their effects by disrupting expression of downstream genes for example? At minimum the site of insertion within each gene, and gene context (potential operon structure) should be described somewhere, perhaps in the supplemental materials.

We thank the reviewer for their concern over polarity and transposon locations. “Both *gldD* (*fjoh_1540*) and *fjoh_0352* are located at the end of short operons (3 and 5 genes, respectively), so we believe polar effects on neighboring genes are unlikely. The transposon in *gldD* is located at base pair (bp) 420 out of a total of 561 bp. We agree that a truncated *gldD* could be produced but are confident in the non-gliding phenotype and, thus, loss of WT GldD function (Fig. 2b). The transposon in *fjoh_0352* is located at base pair 661 bp out of a total of 1074 base pairs, introducing a frameshift and stop codon after 226 amino acids.” Again, even if the truncated protein is not degraded, the function of the mutant protein is not WT based on the altered motility phenotype. We have updated the Methods to reflect this information

- 2) The Shrivastava et al 2012 paper (J. Bacteriol. 194:3678-3688) referenced in the Supplemental section should have been cited in the main text, and discussed in relation to the results observed in the current manuscript (cell aggregate formation and roles of SprB, RemA, and polysaccharides). Each of these were mentioned in the Shrivastava 2012 paper, using some of the same mutants that the authors used. For example, in the 2012 paper, Shrivastava et al also

demonstrated a link of aggregate formation to cell-surface polysaccharides, since mutations that disrupted specific polysaccharide synthesis or secretion genes (*remC*, *wza*, *wzc*) prevented aggregate formation. These points seem relevant to this work on aggregate (*zorb*) formation and movement of these aggregates, and the roles of *RemA*, *SprB*, and of *fjoh_0352* (*wzz* homolog). The results are not duplicative, but they are probably related, and should thus be briefly discussed.

We agree with the reviewer and have moved the Shrivastava 2012 reference to the main text. We have expanded our discussion to include their results and believe it has strengthened our conclusions. Specifically, we have added the following sentences to the results, “Gliding motility adhesins are not only important for surface attachment, but also for cell-cell interactions. Overexpression of *RemA* in *F. johnsoniae* results in strong autoaggregation in liquid culture through interaction with galactose-containing EPS on neighboring cells¹⁷. Whether or not those aggregates are motile is unknown but implicates the importance of gliding adhesins for cell-cell adherence within the microcolony and could explain the abnormal microcolony morphology in the *rema* mutant.

Since *RemA* has been suggested to mediate cell-cell interactions and demonstrates defective microcolony motility, we explored the impact of other physical disruptions in *F. johnsoniae* collective motility.” To further support the hypothesis that disruption of *fjoh_0352* leads to increased cell-cell interactions, we conducted an autoaggregation experiment similar to Shrivastava 2012 and indeed found that *fjoh_0352* aggregates more than both WT and other transposon mutants. The data are presented in Extended Data Fig. 4 in the revised manuscript.

- 3) “previously reported in *F. johnsoniae* (i.e., spinning along an axis with a tethered end)¹⁴” Reference 14 describes this behavior but is not the best reference if only one reference is used, for two reasons. First, Ref 14 used ‘*Cytophaga*’ strain U67. This is probably a member of the genus *Flavobacterium*, but is probably not *F. johnsoniae*. Also, a paper on *Flavobacterium johnsoniae* (then called ‘*Cytophaga johnsonae*’) published 3 years earlier also described this phenomenon. This reference (which corresponds to your current ref 16) is: Pate JL and LYE Chang. 1979. Evidence that gliding motility in prokaryotic cells is driven by rotary assemblies in the cell envelopes. *Curr. Microbiol.* 2:59-64. You could use the Pate reference here, or use both references (Pate and Lapidus).

Agreed. We have updated the manuscript to cite the Pate reference at this location.

- 4) “By contrast, the majority of cells within the zorbs align with random orientation, except for the cells at the base of the zorb, which align perpendicular to the surface.” Clearly, all of the cells at the base do not align perpendicular to the surface. They have many different orientations with respect to that surface; they do not all form a 90 degree angle with respect to that surface. In fact, none of them form a permanent 90 degree angle with respect to the surface (or the zorbs would not be able to move much).

Suggest change to something like:

“By contrast, the majority of cells within the zorbs align with random orientation, except for some at the base of the zorb, which align perpendicular to the surface.”

We agree with the reviewer that the phrase “perpendicular” may have led readers to inaccurate conclusions. We have changed the description of the base cells to either “oblique” or “semi-vertical.”

- 5) Second sentence of discussion, suggest change:
“non-piliated bacterial species”
To
“non-piliated bacterium”

Agreed. The revised manuscript has been updated per the suggestion of the reviewer.

- 6) Near end of discussion suggest change:
“thought to aid *C. canimorsus* traverse host tissue”
to something like
“thought to help *C. canimorsus* traverse host tissues”

We have actually removed any discussion of *C. canimorsus* to focus on *F. johnsoniae*.

- 7) Supplemental Fig 3 title
“Growth dynamics and motility analysis of sprB and remA zorbs.”
It will not be clear to everyone that ‘sprB and remA zorbs’ refers to ‘zorbs of sprB and remA mutants’.
I suggest changing to either
“Growth dynamics and motility analysis of zorbs of sprB and remA mutants.”
Or
“Growth dynamics and motility analysis of sprB and remA mutant zorbs.”

Agreed. The revised manuscript has been updated per the suggestion of the reviewer.

Reviewer #3

1. In the abstract, formulation “...indicates ... by walking on coordinated ... cells” although carefully worded is still an overstatement. The arguments provided by the authors in the manuscript do not necessitate coordination as the only possible explanation of the effect, see point 10 for details.

We accept the feedback of this reviewer and have removed language describing “walking” and “coordination.”

2. In the abstract, last sentence mentioning implications for host pathogen interactions and antibiotic resistance is also too far fetched. First, there is still many things to clarify about the mechanism and second, there is no concrete link to neither of those.

We appreciate this feedback and have revised the last sentence as “This work identifies a new mode of collective movement of bacteria on solid surfaces with potential implications for biofilm dynamics, bacterial cargo transport, microbial adaptation, and other related areas.” In fact, we have removed all reference to *F. johnsoniae* relative and human pathogen *C. canimorsus*. We agree this allows the paper to focus on describing and exploring this novel social biofilm motility.

3. In Fig 1 b, only 3 slices of the z-stack are shown. It would be desired to see full 3D rendering of the images to confirm if the exact shape of the aggregates really matches the sketches by the authors.

We added the full set of slices (8 in total, 6 μm step) to the figure to better display the 3D spherical geometry of the microcolonies as per the suggestion of the reviewer.

4. Authors use ConA labelling to claim the presence of the EPS. To us it does not seem as a sufficient proof that there is EPS present, or how developed it is. In the same context, in the movies, when observing the merging of aggregates one sees their rapid rounding up. Would this fast relaxation to the circular (spherical) shape consistent with the presence of the EPS which is supposed to provide extra mechanical stability to the aggregate? What happens to the EPS during dissolution? What triggers the dissolution process?

Overall, the presence of the EPS is only mentioned in passing in this study, while it certainly deserves more attention. We note, however, that it might be out of scope of this paper.

We thank the reviewer for these interesting and important questions about the nature of EPS in the *F. johnsoniae* microcolonies. It has been an enjoyable task to explore this further and we have included the following sentences in the results and discussion to address the balance between structure and flexibility required for a motile biofilm.

Results: “To determine whether these structures could be biofilms, we stained them with fluorescently-conjugated lectins to detect EPS. Only Concanavalin A (ConA), out of four lectins tested, specifically bound the *F. johnsoniae* microcolonies. As seen from the staining (Fig. 1d,e), ConA binds the core of a microcolony but not individual cells on the surface or planktonic cells, suggesting that EPS production is localized in the cells in microcolonies. Microcolonies bearing multiple, separate EPS cores can be widely observed (Supplementary Movie 2), which is the result of multiple merging events. Indeed, ConA binding only demonstrates the presence of exposed polysaccharides, which could arise from capsular sugars and not biofilm matrix, but the absence of exterior staining of the microcolonies suggests microcolony-specific production and subsequent localization or perhaps a division of labor in which only interior, non-motile cells produce capsule.”

Discussion: “We propose that these microcolonies are biofilms, based on the cohesive multicellular structure and localization of EPS as detected by ConA binding (Fig. 1e, Supplementary Movie 2). In fact, localized ConA staining at the core of a biofilm has been seen previously in *Pseudomonas syringae*, another soil bacterium²⁹, in which ConA binds levan, a branched, fructan polysaccharide. Levan is an intriguing candidate for the matrix of a motile biofilm, as it is water soluble and remains fluid in solution, perhaps contributing flexibility as well as structural integrity^{30,31}. Comparison of annotated levansucrase genes (GO:0050053) from other species of *Flavobacterium* to the *F. johnsoniae* UW101 genome identified *fjoh_2883* (65-69% identity) as a possible levansucrase homolog. A combination of polysaccharide chemical analysis, mutant analysis of *fjoh_2883* and biochemical intervention in WT microcolonies with polymer hydrolyzing enzymes will determine matrix components of the motile *F. johnsoniae* microcolonies. Such studies could illuminate tradeoffs between structural stability, flexibility, and dispersal of *F. johnsoniae* motile microcolonies.”

We are very interested in what triggers the dissolution of the microcolonies but agree it is out of the scope of this paper and hope to explore it in a follow up publication.

5. Line 65. Some advanced image analysis is mentioned in the text, in particular about merging and splitting, growing and shrinking. But in fact no data or analysis on those events is presented - neither in the text, nor in SI.

We direct the reviewer to the Supplementary Movies 1, 4, 5, 7, and 10 for evidence of microcolony growth and merging. In addition, the new Fig. 3, which uses the PMF-inhibitor to show WT microcolony motility is an active process, shows “mature microcolonies quantified at 12 hr and 15 hr display an increase in size concurrent with a reduction of total number over time, demonstrating the ability of the microcolonies to migrate towards one another and merge to form larger, but fewer, microcolonies.” Specifically, the shrinking and splitting events were not a focus of this paper, but we included the possibility as shown in Extended Data Fig. 2 for accurate image analysis.

6. We would like to question if the spider plots and velocity of aggregates are the right (or sufficient) quantities to quantify their motility. The motility of aggregates resembles (active) random motion. This is typically characterised by the mean squared displacement (and other quantities). As the Fig 2 f clearly shows, the velocity of the aggregate is itself a broadly distributed random variable. It would be instructive to see the probability distribution function of those. Authors mention persistence in the text, but this was never quantified. The typical quantities to look in this case are the directional and velocity autocorrelation functions. Thus overall for this point, the statistical analysis of the aggregates’ motility should be significantly expanded.

We agree that aggregate (now referred to as “microcolony”) motion resembles (active) random motion. We didn’t plot the velocity as a frequency histogram because we wanted to show the correlation of the distribution of the microcolony diameter as a function of velocity. Fig. 2f shows the individual microcolonies as points within the boxplots displaying both size and velocity and the median of the whole dataset marked with a cross. Here, the faster microcolonies have a larger median size, which is most clearly seen in *fjoh_0352*. In addition, the shift in the frequency distribution of *fjoh_0352* microcolony velocity is apparent compared to WT.

We contemplated providing further statistical analysis but we have chosen to keep the focus of the analysis on the contribution of gliding with the mutant characterization and computational mechanical simulations of motion (i.e. how single-cell motion can translate to larger-scale motion of the microcolony). Our original use of the word “persistence” was meant to convey the displacement that is shown in Fig. 2c. However, to avoid unintended interpretations that are brought up by this reviewer, we have removed the idea completely from the text. More discussion of this point can be found in response to Comment #10.

7. Fig. 2. What is the reason to use the log scale in d and f if the quantities are changing only about one order of magnitude?

Indeed, the range of the data did not necessarily require a log-scaling; however, “log-scaling was chosen given that the variability/noise in the data (versus measurement uncertainty) scaled with the signal.” For

example, when microcolonies move very slowly, the magnitude of the standard deviation in the velocity is also small, whereas when velocities are high, so are the associated standard deviations. Thus, confidence intervals could be more appropriately calculated and plotted on the log-scale, indicating a fractional uncertainty (e.g. $\pm 10\%$) instead of an absolute magnitude (e.g. $\pm 1 \mu\text{m/s}$). This discussion has been added to Methods for clarity.

8. Line 106: In Fig. a and b, it is very hard to see the circles mentioned by the authors as the “standing” cells. Moreover, that statement seemingly contradicts images in SI, fig 1, which show shorter but still flat lying cells. The statement about “legs” seems to be central for this manuscript. Thus it deserves a better experimental quantification. With a given size of the colonies, confocal or TIRF imaging of the near surface layer of cells at single cell resolution should be feasible. That would clearly demonstrate the fraction and alignment of the cells at the surface.

We appreciate the reviewer for suggesting ways to strengthen the leg (now referred to as “base”) cell orientation argument. We remade Fig. 3 (now Fig. 4) accordingly for a better clarity and guidance to the readers. Specifically: 1) We added the clay model (the old Supplementary Fig. 4) as panel **a** to show the proposed transition in orientation and mode of movement from a single bacterium gliding on substrate to a base cell stroking at the base of a microcolony. 2) We added a threshold-processed, amplified image (now the callout of Fig. 4b) to clearly show the substrate-proximal poles (i.e. the bright dots) of the base cells. We also show how the substrate-proximal poles were quantified in ImageJ in the new Extended Data Fig. 5.

The seeming inconsistency, especially we believe seen on the ctFIRE output images (Extended Data Fig. 1) originates from some of the inherent features of the ctFIRE program. “ctFIRE is a powerful image analysis tool for identifying and quantifying objects in a fiber-shaped morphology (i.e. large aspect ratio). In comparison, ctFIRE doesn’t efficiently capture circular particles or dots, e.g. the substrate-proximal poles of the base cells (again, the bright dots at the base of the microcolony) (Fig. 4b) in this case. Moreover, the default of the software discards short fibers below a preset threshold. Such features of the program left the bright dots not effectively captured on the ctFIRE output images.” This discussion has been added to Methods for clarity.

We tried using confocal in the first place for this purpose but it didn't work “because the stroking movement (at about 2 Hz) is way faster than the scanning speed of the microscope.” So, at least the confocal microscope we had access to was not able to capture the 3D orientation of base cells for an individual time point. Moreover, the WT and the mutants need to be fluorescently labelled for a characterization with fluorescent microscopy, which we plan to do to further characterize the motility mechanism. Given these considerations, we decided to use high-magnification, bright-field microscopy to record and analyze the number of the base cells and the movement. Related discussions have been added to Methods to better clarify and strengthen the rationale of using bright-field microscopy for this purpose.

9. Fig. 3b. What is the rationale for seeing a particular scaling of the number of legs with zorb diameter?

“The intuitive hypothesis (Hypothesis A) upon the relationship between the number of base cells (N) and the microcolony diameter (D) is $N \propto D^2$, in which the surface density of the base cells (n , unit: cells per μm^2) is assumed constant. Another potential relationship (Hypothesis B) between N and D is $N \propto D$. We resolved $n = 0.25$ using the data point of $D = 10 \mu\text{m}$, which translates to a single base cell per $4 \mu\text{m}^2$ of microcolony surface area. For Hypothesis A, $N = n\pi[(\text{Avg. } d/D)D/2]^2 = 0.05D^2$, where $\text{Avg. } d/D = 0.48$ is the averaged ratio between d (diameter of the base of a microcolony) and D measured in experiment (*fjoh_0352* microcolonies at 12 hr after initiating culture, data not shown). For Hypothesis B, $N = nD = 0.25D$. In this case based on Stokes’ law (i.e. $F_{\text{Drag}} = 6\pi\mu Rv$, where F_{Drag} is Stokes’ drag, μ is dynamic viscosity of the fluid, R is the radius of the moving particle, v the particle velocity), the microcolony velocity should remain constant and independent of the size change of the microcolonies (because $v \propto F_{\text{Drag}}/R \propto N/D = \text{Const.}$), which would conflict with data shown in Fig. 2f. We analyzed and plotted the experimentally measured N as a function of D (Fig. 4c) and applied power fitting to get the power-law dependence between N and D . The line on the plot shows the power fitting as $N = 0.2D^{1.36}$, which not only supports the increasing velocity with increasing microcolony size shown in Fig. 2f but also is an important input for the computer simulation. The simulated results between microcolony velocity and microcolony size (Extended Data Fig. 6a) showed a consistent trend compared with the experimental results shown in Fig. 2f.” We added this discussion to Methods to clarify the rationale of the power fitting.

10. One of the points where we would like to strongly disagree and that might be very misleading is about “highly-organised”, “coordinated movement of legs” that leads to achieve “directional persistence”. First, this persistence is not quantified in the manuscript and most probably is based on the visual impression from the movies. Second, and probably most important, an apparent persistence in the movement of aggregates does not imply any “coordination”. As was demonstrated in the context of *Neisseria* aggregates driven by cycles of elongation and retraction of pili filaments, force balance at the level of both, single cells and microcolonies of many cells, may lead to persistent motion without any coordination between stochastic individual pili forces (Pönisch et al 2019). Therefore we believe that the statements of the last paragraph before discussions are misleading and need serious rethinking/reformulation/tests.

We thank the reviewer for pointing out the underdeveloped data presentation and discussion on “the directional persistence” and “coordination”, partially due to the limited space in the original writing. The revised manuscript was switched to the format of a full length article, which allows us to expand the discussion on many essential questions, e.g. identification of the 3D geometry of the microcolonies, evidence of the EPS cores, the motility mechanism, and the simulation method. Specifically in response to this comment: 1) We analyzed and quantified the track displacement of the microcolonies from the WT and the mutants (Fig. 2c). However, those results (the spider plots) got squeezed in a tiny margin in the figure, which impairs the exposure to readers. We reorganized the panels in the figure in the updated manuscript for a better visualization. 2) While direct evidence from the experiment has not been available yet, the conclusion of “coordination” is supported by the distance traveled shown in the track displacement, which we originally described as “directional persistence.” Here, we can understand the example and the discussion brought forwards by the reviewer based on the reference (Pönisch et al 2019). We admit the “directional persistence” was not explicitly defined nor shown in the traditional manner in the original manuscript, which leaves this concept ambiguous and confusing so we have removed it.

Actually, what we were referring to is not only a persistent motion but also, and more importantly, the capability of maintaining the movement in a given direction for a displacement significantly larger than the size of the moving objects (single bacteria in this case). We believe the distance traveled reflected in the spider plots is robust evidence (not direct though) of the coordination (updated to “apparent synchronization rate” in the revised manuscript for better clarity, see details in the next paragraph of this response) among the individual cells at the base of the multicellular microcolony. At a low Reynolds number fluidic environment, the motion of the microcolonies has little influence from inertia, which means there must be a force balance pointing to a constant direction to allow for microcolony displacement. Otherwise, if the force balance kept changing its direction stochastically, there would not be any movement with “directional persistence.” However, to dispel any confusion, we have removed terms related to “directional persistence” from the manuscript. Related discussions have been added to the revised manuscript for a better clarity on the motility mechanism and its experimental evidence.

We have also removed references to base cells being “highly organized” and “coordinated.” Instead, we have clearly defined our goals and analyses in the results section with the following sentences, “Since the *F. johnsoniae* collective movement occurs at a distance significantly larger than the size of single cells (Fig. 2c), the microcolonies, like the strandbeest, must acquire a certain level of synchronization between the movement of base cells. In other words, the net effect of the movement from individual base cells must be a non-zero value, otherwise the composition of the forces from the base cells would be cancelled out, leaving the surface-associated microcolony non-motile. Regardless of the specific translocation mechanism of individual base cells, we aimed to quantify the extent to which base cells synchronize their movement to generate microcolony movement. We define the quantitative variable “apparent synchronization rate” (ASR) as the equivalent percentage of the base cells that synchronize their movement to produce a uniform output of force (i.e. neglecting the heterogeneity of force outputs among individual base cells) on the microcolony. The ASR does not imply the existence of a subgroup of base cells in complete synchronization and the remainder in a complete random movement but rather an “apparent” subset of base cells that are not moving randomly. However, we will refer to these cells as “synchronized” for simplicity.”

11. Concerning the modeling, the authors refer to Ref 18, which, even if a reader tries to go through it, won't really help to understand how the simulations are implemented with respect to this concrete problem of zorbs. Unfortunately there are no further details on this method in the manuscript and thus we can not accept the results of simulations to bare any predictive power.

We thank the reviewer for the opportunity to provide more details of the simulation. We have added more discussion in the revised manuscript [two paragraphs in Results of the main text, and a whole section in SI including (i) Governing equations in the computer simulation, (ii) Numerical method, and (iii) Validation of the numerical method along with Supplementary Fig.1] about why/how we use this specific numerical method (GMLS) to model the problem studied in this work. Briefly, the governing equations were established based on a set of five premises outlined in the text but copied here:

- (i) The dynamics of the system lies in the low Reynolds number regime ($Re \approx 1e-6$) considering the small inertia effect produced by the small length scale and velocity of the bacteria and microcolonies. Therefore, the propulsion force is always balanced with the Stokes' drag (F_{Drag}) exerted by the fluid.

- (ii) The simulation model was reduced to two-dimensional (2D) since the movements of bacteria and microcolonies are all surface-associated.
- (iii) The under-oil microdrop is treated as an incompressible fluid with a finite boundary.
- (iv) The possible rotation of the bacteria or microcolonies is negligible and hence not considered in the simulation.
- (v) All the input data used in the simulation was from the experimental results of the hyper-motile mutant *fjoh_0352*.

Then, the numerical method which was chosen to solve the governing equations was described along with supporting reasons. We also added a benchmark to validate the accuracy of the numerical method (Supplementary Fig. 1). Finally, we built the computational model of the problem studied in this work based on the input data obtained from the experiment and ran the simulation to quantify the “apparent synchronization rate” (ASR) of the base cells.

12. Not found in the review.

13. While the pinwheel motion of tethered cells sounds like a plausible mechanism of force generation, most of the aspects of how that microscopic force generation results in the macroscopic motion of the colonies is not really explained, and we dare to say not understood. How individual force-dipoles tethered to the zorb propel the whole zorb is not explained or modeled. If there is a coordination between individual “legs” how is that achieved? When they make a stroke - are they synchronised? are they spatially correlated? All those, in principle, are central to understanding the mechanism and can be experimentally accessed and quantified.

We appreciate the feedback from the reviewer about the necessity to clarify our proposed mechanism. To do so, we have included in the discussion a comparison of multiple possible mechanisms, how our data supports the proposed mechanism, and what steps we would take next to test our hypothesis:

“How the activity of gliding apparatus from individual *F. johnsoniae* cells translocates to the movement of an entire microcolony remains unknown. The movement of the microcolonies via base cells briefly gliding when attached to the surface has not been conclusively excluded. Additionally, the many base cells may act as focal adhesions successively binding to the substrate, not unlike the pili used by *Neisseria*. However, due to the observations of pinwheeling and the dynamic nature of the cells within the microcolony, we propose the hypothesis of a “power stroke” force originating from base cells tethered to the microcolony (not surface) by their rotary gliding motors that propel the microcolony forward upon contact with the surface. Regardless of mechanism, the computational simulations demonstrate the extent to which intercellular synchronization is required among the base cells that leads to translocation of the microcolony on a millimeter scale. Future experiments with fluorescent WT cells to track behavior of single cells within the microcolony, especially at the base, will begin to differentiate among the possible mechanisms.”

As explained in Fig. 4 and Extended Data Fig. 6, we do believe there is a partial synchronization between the base cells, which we have clarified in the results section and have described in detail in the response to Comment #10. Briefly, in response to this comment, we included the following sentences, “We define the

quantitative variable “apparent synchronization rate” (ASR) as the equivalent percentage of the base cells that synchronize their movement to produce a uniform output of force (i.e. neglecting the heterogeneity of force outputs among individual base cells) on the microcolony. The ASR does not imply the existence of a subgroup of base cells in complete synchronization and the remainder in a complete random movement but rather an “apparent” subset of base cells that are not moving randomly.” As well as, “Whether or not the apparent synchronization of base cells occurs randomly or is subject to regulation remains unknown.” We completely agree that probing further how base cells coordinate movement is important for our full understanding of the mechanism of microcolony motility but suggest it is out of the scope of the current manuscript.

14. On the movies of aggregation process it is easy to notice events of aggregates being attracted to each other before they even touch. What is the mechanism of that (and certainly that is also related to the motility mechanism). What are the forces driving the merging of aggregates?

We totally agree that the attraction between two microcolonies during the merging process is a very interesting phenomenon, which is worth further study on the molecular mechanism. For example, a systemic examination of the metabolites at different stages of microcolony development could shed light on regulation. We believe it would be a great topic for a follow-up paper exploring the merging process and the potential for regulating microcolony directionality, which could be related.

15. In the Discussion, concerning the movement of colonies, *P.aeruginosa* is another well known example (Sauer et al 2002).

We agree with the reviewer that *P. aeruginosa* microcolonies deserve to be included. We also received feedback that *Myxococcus* is a good example for social biofilm. Thus, we have included the following sentences in the introduction, which we believe highlights the novelty of social biofilm motility dependent on gliding motility, “Small *Pseudomonas aeruginosa* microcolonies are dynamic, using type-IV pili to merge and split before settling down to form a sessile biofilm⁹. *Myxococcus xanthus* displays diverse multicellular and motile behaviors. Individual cells rely on gliding motility whereas social motility for bacterial predation and the formation of fruiting bodies primarily depends upon type-IV pili⁶. *Neisseria* species also use type-IV pili to form large microcolonies that move across a surface as discrete units and grow via merging^{10,11}. In these examples, type-IV pili are important contributors to the movement of microcolonies.”

REVIEWER COMMENTS

Reviewer #1 (Remarks to the Author):

Thanks to the authors for taking my comments into such careful consideration. I think the RMSD comparison to aggregate size does a nice job of eliminating the Brownian adhesion model. Nice! I also think the PMF disruptor is a fine substitute for the motor mutant. Thank you!

About the citations related to *M. xanthus*, IIRC a *pilA* mutant still forms fruiting bodies. See if you can support the statement that pili are required for fruiting body formation rather than rely on the review reference 6 because I don't think it is true. Thus, while it "makes sense" social motility" would be important for fruiting body formation, I seem to recall that pilus mutants have EPS defects that account for the fruiting defect. See Yang, Lux, Hu, Hu, Shi 2010. PilA localization affects extracellular polysaccharide production and fruiting body formation in *M. xanthus*. *Mol Micro* 76:1500-1513.

As point of interest, lab strain *M. xanthus* are mutants with reduced aggregation and wild isolates grow as large hollow spheres (3mm-ish diameter IIRC) that roll around an agitated flask.

Moreover, they tend to have holes (openings) in the sphere and I seem to recall old literature showing that gliding motility could be used to import long filaments of cyanobacteria which would be digested inside the sphere. Hyper weird and not sure if it is biologically relevant (don't know if they form these spheres in nature) but it changed the way I thought about *M. xanthus* and your paper reminded me of this. Note: I looked to find references and although I recall something in *Science/Nature* about it in the 50s/60s this is what I found: Burnham, Collart, Highison 1981.

Entrapment and lysis of the cyanobacterium *Phormidium luridum* by aqueous colonies of *Myxococcus xanthus* PCO2. *Arch Microbiol.* 129:285-294. Perhaps refs therein. You don't need to include this or anything, might be helpful/interesting down the line.

In the absence of knowing where or how force is generated, the authors might consider taking a more agnostic approach by acknowledging that force could be generated at the base or tip. This way they aren't favoring one model over another (base vs tip or both) in the future.

Dan Kearns

Reviewer #3 (Remarks to the Author):

The revised version of the manuscript has improved on almost all levels (there is however, one serious point that could and should be fixed). This time I focused on reading the paper as if it was a new submission and to check if that makes a balanced and interesting statement. In my opinion it does, although it doesn't explain the mechanisms, but it opens up a very exciting topic of research where many interesting things can be learned in the future. And thus I think it is good for *Nature Communications*.

I do have one serious "but" and it concerns the model and the mentioning of the synchronization. The authors would either have to answer that comment or remove or strongly rephrase the corresponding parts in the text. In its current form it is really wrong, I think.

Now the model is explained really to the point where I finally could understand what it is about and that also opened up the related problem. In fact what the authors do is to use a very advanced modeling approach to solve the Navier-Stokes equation assuming that a force propels either an individual cell surrounded by a bunch of others or a bigger blob. The propulsion force is balanced by hydrodynamic friction and that's it. However, it is really like using a cannon against a fly. If the authors wanted to estimate the force by using the Stokes friction law it could be done just as an estimate: $F = 6\pi\eta R v = 6 * 3 * 8 * 10^{-4} \text{ Pa s} * 0.3 * 10^{-6} \text{ m} * 2 * 10^{-6} \text{ m/s} = 8,6 * 10^{-15}$ which is of the same order of magnitude as a much more complex calculation by the authors. However, that is not the main problem. The main problem is to assume that the gliding motility force can be calculated from the velocity argument. In fact in the literature one can find the estimates of the so-called stalling force of the gliding motility which for different species mounts up to tens to hundreds of piconewtons, whereas the estimate from the friction law is 10^{-15}

3 piconewtons! 5 orders of magnitude smaller. Similarly, pili driven motility generates also hundreds of piconewtons. Thus what I want to say is that the bacteria are able to generate much larger force before they are stalled and stop moving. Correspondingly - calculating the total force required to move the whole colony assuming that it is only resisted by friction and arriving to a number which is still a fraction of a piconewton, and then claiming that for that one needs to coordinate several cells is simply wrong. If anything that confirms the opposite - there is no synchronisation. One cell would be able to pull the whole colony if that would be the friction resisting its motion.

In fact what is happening is that multiple cells are engaging in force generation at random, where every individual cell may generate the force at the level of piconewtons (so 3 orders of magnitude larger than what is written in the paper) but as they compete with each other in mechanisms of tug of war that was many times invoked in the context of molecular motor and pili-driven motility, their collective but random operation results in the overall random motion of the colony, without any coordination. Thus the word synchronisation should be removed from the paper. Also here, the effect of the increasing mobility with increasing size can be explained by the random nature of individual forces. As the number of random force generation units increases, so do the fluctuations (famous $N^{1/2}$ law), and as it is the fluctuation that drives the motion of the colony, also its mobility is increasing.

Thus my suggestion to remove the current estimates from the paper as they contradict published literature on the scale of the gliding motility forces and lead to a misleading statement of the synchronisation. The authors could focus on the novelty of the observed effect and admit, as they do now in the discussion, that the understanding of the exact mechanism requires further work.

I would be happy to recommend this paper for publication if the above problem would amended in the revised version.

To All Reviewers:

We thank the reviewers for their time and effort in reviewing the resubmission and appreciate their affirmation regarding the value of this work. We carefully addressed the additional concerns from the reviewers and further updated the manuscript accordingly. Below we provide a point-by-point response to the reviewers' comments. **Response to reviewer comments shown in "red".**

Reviewer #1

Thanks to the authors for taking my comments into such careful consideration. I think the RMSD comparison to aggregate size does a nice job of eliminating the Brownian adhesion model. Nice! I also think the PMF disruptor is a fine substitute for the motor mutant. Thank you!

We highly appreciate these encouraging words. It energizes us to keep exploring this type of unique collective movement found in gliding bacteria.

About the citations related to *M. xanthus*, IIRC a *pilA* mutant still forms fruiting bodies. See if you can support the statement that pili are required for fruiting body formation rather than rely on the review reference 6 because I don't think it is true. Thus, while it "makes sense" social motility" would be important for fruiting body formation, I seem to recall that pilus mutants have EPS defects that account for the fruiting defect. See Yang, Lux, Hu, Hu, Shi 2010. PilA localization affects extracellular polysaccharide production and fruiting body formation in *M. xanthus*. *Mol Micro* 76:1500-1513.

Agreed. We have updated the references to include primary literature showing the original 1979 observation of two separate motility systems and a paper that shows *pilA* mutants form defective fruiting bodies. The EPS phenotype you reference is also necessary for fruiting body formation as it is thought to be the binding site of Myxo type-IV pili! <https://doi.org/10.1073/pnas.0836639100>.

As point of interest, lab strain *M. xanthus* are mutants with reduced aggregation and wild isolates grow as large hollow spheres (3mm-ish diameter IIRC) that roll around an agitated flask. Moreover, they tend to have holes (openings) in the sphere and I seem to recall old literature showing that gliding motility could be used to import long filaments of cyanobacteria which would be digested inside the sphere. Hyper weird and not sure if it is biologically relevant (don't know if they form these spheres in nature) but it changed the way I thought about *M. xanthus* and your paper reminded me of this. Note: I looked to find references and although I recall something in *Science/Nature* about it in the 50s/60s this is what I found: Burnham, Collart, Highison 1981. Entrapment and lysis of the cyanobacterium *Phormidium luridum* by aqueous colonies of *Myxococcus xanthus* PCO2. *Arch Microbiol.* 129:285-294. Perhaps refs therein. You don't need to include this or anything, might be helpful/interesting down the line.

We thank the reviewer for this information. The image of an "aqueous colony" is really striking and certainly a pertinent comparison. The paper describes a "tightly-woven mass" of exterior cells, which is an interesting contrast to the *Flavobacterium* microcolonies. The function of the aqueous colonies as predatory, or at least nutritionally motivated, is a good place to start exploring the potential function and features of the microcolonies.

In the absence of knowing where or how force is generated, the authors might consider taking a more agnostic approach by acknowledging that force could be generated at the base or tip. This way they aren't favoring one model over another (base vs tip or both) in the future.

We totally agree with this suggestion. Since we have not conducted the experiments to differentiate mechanism and cellular function within a microcolony, it makes strategic sense to be more agnostic. We adjusted text in the main body to reflect the change and encourage further studies in the future.

Reviewer #3

The revised version of the manuscript has improved on almost all levels (there is however, one serious point that could and should be fixed). This time I focused on reading the paper as if it was a new submission and to check if that makes a balanced and interesting statement. In my opinion it does, although it doesn't explain the mechanisms, but it opens up a very exciting topic of research where many interesting things can be learned in the future. And thus I think it is good for Nature Communications.

We are excited to see such positive and encouraging feedback from the resubmission.

I do have one serious "but" and it concerns the model and the mentioning of the synchronization. The authors would either have to answer that comment or remove or strongly rephrase the corresponding parts in the text. In its current form it is really wrong, I think.

In general we totally understand and agree with this concern. After a careful and detailed discussion among our primary authors upon the questions asked below, we had the mechanism parts in the text carefully rephrased. As mentioned and suggested by the reviewers, we first further clarified the rationale of the simulation, assumptions made, and removed the word "synchronization".

Now the model is explained really to the point where I finally could understand what it is about and that also opened up the related problem. In fact what the authors do is to use a very advanced modeling approach to solve the Navier-Stokes equation assuming that a force propels either an individual cell surrounded by a bunch of others or a bigger blob. The propulsion force is balanced by hydrodynamic friction and that's it. However, it is really like using a cannon against a fly. If the authors wanted to estimate the force by using the Stokes friction law it could be done just as an estimate: $F=6\pi\eta R v = 6 * 3 * 8 * 10^{-4} \text{ Pa s} * 0.3 * 10^{-6} \text{ m} * 2 * 10^{-6} \text{ m/s} = 8,6 * 10^{-15}$ which is of the same order of magnitude as a much more complex calculation by the authors.

First of all, thanks for the reviewer's recognition of our advanced modelling approach. We actually solved the Stokes equation but not the full Navier-Stokes equation due to the low Reynolds number ($Re \approx 1e-6$) of the problem. However, the Stokes law is valid only if we consider an isolated object moving in an unbounded (infinitely large) fluid environment; otherwise, the boundary effect and the hydrodynamic interaction between the object and other surrounding objects (e.g. the relative motion between the moving objects) must be considered, and hence the estimation from the Stokes law in this case would be inaccurate. We further clarified the rationale of using an advanced simulation to understand the collective movement and the driving force as follows:

i) For individual bacteria, since the *F. johnsoniae* bacteria in a local niche typically move parallelly in one direction (this phenomenon on *F. johnsoniae* has been widely reported in literatures and observed in our experiments), the drag force from the simulation is 4 times smaller than is estimated by the Stokes law (Ref. [1], the equation to estimate the gliding force of *F. johnsoniae* from the translational friction drag: $F = 8\pi\eta av / (\ln 2a/b + 0.5)$, where a is the half length of the bacterium, b is the half width of the bacterium, v is its translational velocity, and η is the fluid viscosity. So, $F = 8 * 3.14 * 1^{-3} * 3.37^{-6} * 2^{-6} / (\ln 2 * 3.37 / 0.36 + 0.5) \approx 24.2^{-15} \text{ N}$).

ii) In the microcolony simulation, since the object behavior of the microcolonies (e.g. the size or the relative moving directions among the microcolonies) is different from single cells, the drag force obtained from the simulation is 5 times larger than is estimated by the Stokes law (Ref. [2], the equation to estimate the propulsion force on a microcolony with diameter (D) of 50 μm : $F = 4\pi\eta Dv\epsilon(1 - \epsilon^2)$, where $\epsilon = 1/[-0.077216 - \ln(\text{Re}/4)]$, D is the diameter of the microcolony, v is its translational velocity, and η is the fluid viscosity. So, $F = 4 \cdot 3.14 \cdot 10^{-3} \cdot 50 \cdot 10^{-6} \cdot 0.031 \cdot 10^{-6} \cdot 0.066 \cdot (1 - 0.066 \cdot 0.066) \approx 1.3 \cdot 10^{-15}$ N, where $\epsilon = 1/[-0.077216 - \ln(10^{-6}/4)] \approx 0.066$).

iii) The underoil microdrop is not an unbounded fluid environment. The oil:water:surface interface causes increased pressure and viscous stress within the microdrop.

All these deviations make the Stokes law highly limited in solving and understanding a real-world problem. The modeling approach used in this work enables substantially improved accuracy and precision compared to the Stokes law, which is essential and necessary if one tries to acquire visions from such complex fluid-object interactions. We believe the simulations deployed to model the relationship between base cells is mathematically sound and an advance in the field of computational biology by increasing the precision of the Stokes equation under complex fluidic environments.

However, that is not the main problem. The main problem is to assume that the gliding motility force can be calculated from the velocity argument. In fact in the literature one can find the estimates of the so-called stalling force of the gliding motility which for different species mounts up to tens to hundreds of piconewtons, whereas the estimate from the friction law is 10^{-3} piconewtons! 5 orders of magnitude smaller. Similarly, pili driven motility generates also hundreds of piconewtons. Thus what I want to say is that the bacteria are able to generate much larger force before they are stalled and stop moving. Correspondingly - calculating the total force required to move the whole colony assuming that it is only resisted by friction and arriving to a number which is still a fraction of a piconewton, and then claiming that for that one needs to coordinate several cells is simply wrong. If anything that confirms the opposite - there is no synchronisation. One cell would be able to pull the whole colony if that would be the friction resisting its motion.

Overall, reading the reviewer's comments and aligning them with our presentation of data in the paper, we believe that an explanation for the disconnect is the idea that the model attempts to calculate the force generated at the substrate-proximal pole of a base cell with a tethered rotary motor at its microcolony-proximal pole (see the last schematic in Fig. 4a), as that is the mechanism of translocation most discussed in this work. Further confusion may have arisen from the strandbeest analogy, which has been removed. Specifically, we want to respectfully point out that the force (or torque) generated by a gliding motor is not necessarily 100% converted to (or reflected by) the gliding force (or traction) required to lead to and maintain the velocity of the bacteria observed in experiment. This point, by itself, could be a very interesting research direction in biophysics. Rather, the focus of the simulation in this work is to get a calculation upon the drag force of each "perpendicular" base cell at the substrate-proximal pole based on geometry of surface contact and a further comparison between the drag force generated by a single base cell and the drag force required to propel a microcolony with a given size and velocity, agnostic of the actual mechanism and balancing the cumulative forces of base cells with the drag force on the microcolony. To the best of our knowledge, the gliding force in *Flavobacterium* has not been experimentally quantified but a previous study computationally estimated a gliding force of $24.2 \cdot 10^{-15}$ N (see Ref. [1] and the discussion above), which is on par with our calculation but about 4 times higher due to the lack of accuracy of the equation used in that work compared to our simulation method. In addition,

we presume the stalling force mentioned by the reviewer was from *Neisseria* or *Myxococcus* that both use type IV pili to glide, which we agree deploys large forces (i.e. tens to hundreds of piconewtons) during retraction of the pili (Ref. [3]). Interestingly, *Myxococcus* does use a true gliding rotary mechanism as well, but the traction (or the gliding force) of which on the surface is hardly measurable (Ref. [4]) compared to its pili retraction mechanism. Overall we agree that the description of the assumptions and limitations were not adequately covered in the manuscript and we thank the reviewer for this crucial feedback. We have updated the results to reflect these points.

In fact what is happening is that multiple cells are engaging in force generation at random, where every individual cell may generate the force at the level of piconewtons (so 3 orders of magnitude larger than what is written in the paper) but as they compete with each other in mechanisms of tug of war that was many times invoked in the context of molecular motor and pili-driven motility, their collective but random operation results in the overall random motion of the colony, without any coordination. Thus the word synchronisation should be removed from the paper. Also here, the effect of the increasing mobility with increasing size can be explained by the random nature of individual forces. As the number of random force generation units increases, so do the fluctuations (famous $N^{1/2}$ law), and as it is the fluctuation that drives the motion of the colony, also its mobility is increasing.

We agree that random force generation of the base cells cannot be excluded and we have removed any reference to the word “synchronization” from the abstract, results and discussion as the reviewer has suggested. However, we did calculate the predicted net percentage of base cells that exert a force to generate the experimentally quantified velocity of 50 μM diameter microcolonies. We recognize that synchronization holds a certain higher-level meaning of orchestration so we have changed the term to the “net force percentage” or “NFP.” We have rewritten this part of the results section to focus only on force generation and avoid implying base cells are purposefully regulated in their movements.

Thus my suggestion to remove the current estimates from the paper as they contradict published literature on the scale of the gliding motility forces and lead to a misleading statement of the synchronisation. The authors could focus on the novelty of the observed effect and admit, as they do now in the discussion, that the understanding of the exact mechanism requires further work.

I would be happy to recommend this paper for publication if the above problem would amended in the revised version.

We greatly appreciate the comments and suggestions from the reviewer and we believe it helps us further enhance the rigor of the discussion about force generation in this study. As discussed previously, we agree the “synchronization hypothesis”, in its current state, is in lack of direct and sufficient evidence from experiments and has been removed. Overall, the simulation and the mechanism parts were all carefully revised according to the discussions listed above.

References

1. Abhishek Shrivastava, Pushkar P. Lele, and Howard C. Berg. “A rotary motor drives *Flavobacterium* gliding” *Curr. Biol.* **25**, 338-341 (2015).
2. Burke Huner, and R. G. Hussey. “Cylinder drag at low Reynolds number” *Phys. Fluids* **20**, 1211-1218 (1977).
3. Berenike Maier, Laura Potter, Magdalene So, Hank S. Seifert, and Michael P. Sheetz. “Single pilus motor forces exceed 100 pN” *Proc. Natl. Acad. Sci. USA.* **99**, 16012-16017 (2002).
4. Benedikt Sabass, Matthias D. Koch, Guannan Liu, Howard A. Stone, and Joshua W. Shaevitz. “Force generation by groups of migrating bacteria” *Proc. Natl. Acad. Sci. USA.* **114**, 7266–7271 (2017).

REVIEWERS' COMMENTS

Reviewer #3 (Remarks to the Author):

In general, authors adjusted the wording in the way to keep the language consistent with what data shows, thus I welcome these changes and recommend the paper for publication.

However, I want to still provide a small comment, which might yet motivate the authors to think of how they present results in the final version of the manuscript.

In the ref [1] mentioned in the reply, I couldn't find the corresponding calculation of the gliding force. They only seem to mention torques. However, the torques themselves in that paper are estimated from the friction force. Thus it is a circular argument. The point here being that it is highly likely, or not a priori impossible to assume that both rotational and translational forces generated by individual cells are orders of magnitude larger than those estimated solely on friction argument. There are limitations in speed for most known molecular motors (eukaryotic or bacteria) but they are not due to friction. And stalling forces are typically in the piconewtons range. Thus I only want to caution from making this possibly false statement. Or at least mention that it is the assumption taken in the manuscript (estimate individual cell forces as being equal to drag force) that requires clarification in the future.

REVIEWERS' COMMENTS

Reviewer #3 (Remarks to the Author):

In general, authors adjusted the wording in the way to keep the language consistent with what data shows, thus I welcome these changes and recommend the paper for publication.

However, I want to still provide a small comment, which might yet motivate the authors to think of how they present results in the final version of the manuscript.

In the ref [1] mentioned in the reply, I couldn't find the corresponding calculation of the gliding force. They only seem to mention torques. However, the torques themselves in that paper are estimated from the friction force. Thus it is a circular argument. The point here being that it is highly likely, or not a priori impossible to assume that both rotational and translational forces generated by individual cells are orders of magnitude larger than those estimated solely on friction argument. There are limitations in speed for most known molecular motors (eukaryotic or bacteria) but they are not due to friction. And stalling forces are typically in the piconewtons range. Thus I only want to caution from making this possibly false statement. Or at least mention that it is the assumption taken in the manuscript (estimate individual cell forces as being equal to drag force) that requires clarification in the future.

We thank the reviewer for the further discussion on the gliding force. We completely agreed and as expressed in the previous response, “To the best of our knowledge, the gliding force in *Flavobacterium* has not been experimentally quantified but a previous study computationally estimated a gliding force of 24.2×10^{-15} N”. To further articulate this fact and avoid conveying any misleading guidance, we agreed with the reviewer’s suggestion and added a sentence to Discussion as “It is our assumption that individual cell forces are equal to the drag force, which requires further clarification in the future.”.